# LCA-ON-THE-LINE: BENCHMARKING OUT OF DISTRIBUTION GENERALIZATION WITH CLASS TAXONOMIES

## ABSTRACT

In this paper, we address the challenge of predicting models' Out-of-Distribution (OOD) performance from in-distribution measurement. We found that prior evaluations, notably Miller et al. (2021) Baek et al. (2022), become less robust in comparing models pretrained on different dataset, or Vision-only vs Vision-language models on significant distribution shift datasets(like ObjectNet Barbu et al. (2019)). In this work, we reintroduce the Least Common Ancestor (LCA) distance, a metric that has been largely overshadowed since ImageNet Challenge. By leveraging class hierarchy like WordNet, we utilize the LCA to measure the taxonomic distance between labels and predictions, presenting it as a benchmark for model generalization. On 75 models spanning five severe shifted ImageNet-OOD datasets, we proven LCA is especially robust among models of different settings by revealing a strong linear correlation between in-domain ImageNet LCA scores and OOD Top1 accuracy across ImageNet-S/R/A/ObjectNet. This discovery gives rise to a novel evaluation framework termed 'LCA-on-the-Line', facilitating unified and consistent assessments across a broad spectrum of models and datasets. This benchmark might help explaining the surprising results that zero-shot vision-language models with poor top-1 accuracy generalize better to novel datasets compared to state-of-the-art vision models.

Besides introducing an evaluative tool, we also delve into the intricate ties between the LCA metric and model generalization. By aligning model predictions more closely with the WordNet hierarchy and refining prompt engineering in zero-shot vision-language models, we offer tangible strategies to improve model generalization. We challenge the prevailing notion that LCA offers no added evaluative value over top-1 accuracy, our research provides invaluable insights and actionable techniques to enhance model robustness and generalization across various tasks and scenarios.

## 1 INTRODUCTION

Generalizing models trained on in-distribution (ID) data to out-of-distribution (OOD) conditions is a notoriously challenging task. This is primarily because distribution shifts can undermine the IID assumption between training and testing data, thereby affecting robust performance. Work from OOD detection have target shrift in distribution by identifying anomalies (Sun et al., 2021; Ren et al., 2021; Liang et al., 2018; Liu et al., 2020). Besides, numerous OOD datasets have been proposed to study the effects of different interventions, such as temporal shifts (Hu et al., 2022; Lomonaco & Maltoni, 2017; Lin et al., 2021), artificial noise (Hendrycks & Dietterich, 2019; Arjovsky et al., 2019; Larochelle et al., 2008), and natural distribution shifts (Hendrycks et al., 2021; Hendrycks & Dietterich, 2019; Barbu et al., 2019; Recht et al., 2019). Notably, the challenge of maintaining model robustness becomes significantly more difficult with severe visual shifts in the image domain.

**Estimating OOD Generalization**: Within the sphere of model generalization, numerous attempts, following the concept of *effective robustness* (Taori et al., 2020), have been made to estimate a model's performance on OOD datasets based on in-domain measurements(Fig 1). These approaches have been referred to as 'XX-on-the-line'(Miller et al., 2021; Baek et al., 2022), which involve modeling correlations of OOD performance with in-domain accuracy (Miller et al., 2021; Recht et al., 2019; Miller et al., 2020; Roelofs et al., 2019) or models consensus on in-domain accuracy (Jiang et al., 2021; Baek et al., 2022).

In prior attempts, several methods rely on domain generalization strategies that necessitate prior knowledge of the target domain or require an estimation of OOD domain information (Chen et al., 2021; Li et al., 2022a). These can lead to computationally intensive processes in practice, particularly when involving multiple models or inferences (Baek et al., 2022; Deng et al., 2022).

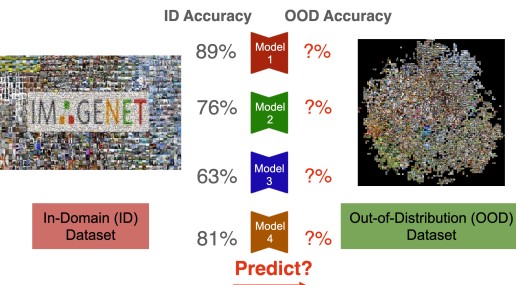

Figure 1: We focus on evaluating how well models generalize to unseen, out-of-distribution (OOD) datasets. Specifically, we aim to predict a model's OOD performance, based on its performance in a familiar, in-domain setting.

Furthermore, many of these studies target generalization on OOD datasets with limited visual shifts or only involved artificial noise, such as ImageNet-v2 or ImageNet-C (Recht et al., 2019; Arjovsky et al., 2019). Such datasets fail to reflect a model's generalization capability when confronted with severe distribution shifts(Hendrycks et al., 2021; Hendrycks & Dietterich, 2019; Barbu et al., 2019), as there is often limited transfer of robustness from synthetic to natural distribution shifts (Taori et al., 2020).

Moreover, most prior researches has focused solely on evaluating supervised vision-only models trained on ImageNet (Taori et al., 2020; Mustafa et al., 2020). However, the rise of large-scale language models trained on dataset like LAION, particularly given their impressive performance in robust OOD generalization, underscores the necessity to evaluate and compare models across different families under a unified evaluation framework.

Unlike Vision Models (VMs), VLMs leverage more diverse training data, contrastive base loss, and language supervision. There have been prior attempts to solely measure VLM generalization (HaoChen et al., 2021; Fang et al., 2022; Schuhmann et al., 2022; Kaur et al., 2022), specifically, training data diversity has been suggested as an indicator of model generalization, but collecting or training on such extensive data can be non-trivial (Schuhmann et al., 2022). Among prior attempts, a unified, simple benchmark between both VLMs and VMs that can explain model generalization and be converted into actionable improvements is still lacking.

In light of this, it is essential to establish a unified benchmarking metric robustly applicable across both VMs and VLMs, to assess model generalization. Our experiment observed that prior art, like accuracy-on-the-line(Miller et al., 2021), fail to explain the increment on effective robustness from VLMs to VMs. Recently, (Shi et al., 2023) have observed the same problem and propose to evaluating OOD accuracy using multiple ID test sets, but still required multiple evaluation.

To address these issues, we propose to adopt Least Common Ancestor (LCA) score, to measure model generalization. LCA distance is the taxonomic distance between labels and predictions, given a predefined class hierarchy, such as WordNet. Through a series of empirical experiments involving 75 models of different modalities (36 VMs and 39 VLMs), we show, for the first time to our knowledge, that the in-domain LCA metric **strongly correlates** with multiple ImageNet-OOD datasets under severe visual shifts (ImageNet-Rendition (Hendrycks et al., 2021), Sketch (Hendrycks & Dietterich, 2019), Adversarial (Hendrycks et al., 2021), and ObjectNet (Barbu et al., 2019)). This finding may help explain the surprising result that zero-shot vision-language models with poor top-1 accuracy generalize better to novel datasets compared to state-of-the-art vision models, which spurs us to further investigate and discuss the potential of the LCA benchmark for improving model generalization. **Please refer to section 3 for our motivation and hypothesis of adopting LCA, and settings comparison to prior work are illustrate in Fig 8**.

In summary, this paper contributes the following:

**(1).** We propose a novel benchmark, the Least Common Ancestor (LCA) distance, to assess model generalization. This approach utilizes the class hierarchy like WordNet to encode interclass relationships. **(2).** We perform large-scale experiments to validate our proposed benchmarking strategy. We empirically study 75 models across five ImageNet-OOD datasets, showcasing a strong linear relationship between in-domain LCA and OOD Top1 performance across models with different configurations, establishing an 'LCA-on-the-Line' framework. **(3).** We provide a detailed analysis and discussion of the underlying connection between the LCA and model generalization, offering fresh insights to stimulate future work. **(4).** We demonstrate the potential usage of this benchmark by

showing how model generalization can be improved by aligning model predictions with the WordNet hierarchy.

## 2 LCA DISTANCE AND ELCA DISTANCE MEASURE MISTAKE SEVERITY

We propose the use of in-domain Lowest Common Ancestor (LCA) distance, or taxonomy loss, as a benchmark for model generalization. Here, we will formally define how taxonomy loss can be measured using in-domain data.

Taxonomy loss measure the distance between model's prediction of each class likelihood, to a predefined class order encoded by class taxonomy. Lower loss expect model to 'make better mistake' Bertinetto et al. (2020), by assigning higher likelihood to class that is semantically closer to the ground truth class. Following previous research (Bertinetto et al., 2020; Deng et al., 2009a), we utilize WordNet (Miller et al., 1990), a large-scale lexical database inspired by psycholinguistic theories of human lexical memory (Miller, 1995), to encode class taxonomy. A example of LCA distance is shown in Fig 2.

Given two classes, $y$ (the ground truth class) and $y'$, we define the **LCA distance** according to (Bertinetto et al., 2020) as $lcad(y', y) := f(y) - f(lca(y, y'))$, where $f(y) \geq f(lca(y, y'))$ and $lca((y', y))$ denotes the lowest common ancestor of nodes $y$ and $y'$ within the predefined Word-

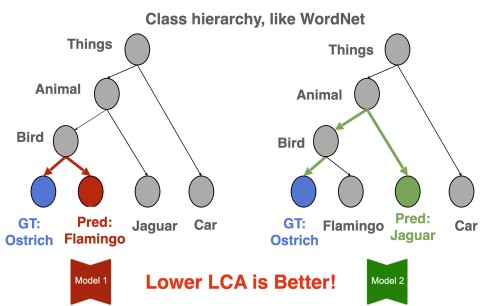

Figure 2: Our method involves measuring a model's generalization based on its in-domain semantic severity of mistake. We use the 'Least Common Ancestor' (LCA) distance, which is the distance between the model's prediction and the ground truth class in a predefined taxonomy hierarchy, like WordNet. LCA distance is ratio to shortest path from prediction to ground truth class in hierarchy tree

Net hierarchy, and $f(\cdot)$ represents a function of a node, such as the tree depth. We use the information content as described in (Valmadre, 2022).

For each sample $X_i$ in the given dataset $\mathcal{D} := X_1, \ldots, X_n$: $LCAD(model, \mathcal{D}) := \frac{1}{n} \sum_{i=1}^{n} lcad(\widehat{y_i}, y_i) \iff y_i \neq \widehat{y_i}$, where $\widehat{y_i}$ is the predicted class for sample $X_i$ using the model, $y_i$ is the true class for sample $X_i$, and $y_i \neq \widehat{y_i}$. Intuitively, a model with a lower LCA distance demonstrates greater semantic understanding on class ontology in WordNet.

We can also generalize LCA distance to settings where the model outputs a distribution over all possible classes for each sample (like using softmax). For a sample $X_i$ whose ground truth class is $y_i$, and the model outputs $(\widehat{p}_{i,1}, \ldots, \widehat{p}_{K,1})$ over the $K$ classes (e.g., 1000 in ImageNet), we define **Expected Lowest Common Ancestor Distance (ELCAD)**: $ELCAD(model, \mathcal{D}) := \frac{1}{nK} \sum_{i=1}^{n} \sum_{k=1}^{K} \widehat{p}_{k,i} \cdot lcad(k, y_i)$. From a probabilistic perspective, ELCAD is a weighted measure of mistake severity according to the model's confidence on each node in hierarchy. Intuitively, it combine LCA distance with cross entropy measurement.

| Model | ImageNet | | | ImageNetv2 | | | ImageNet-S | | | ImageNet-R | | | ImageNet-A | | | ObjectNet | | |
|---|---|---|---|---|---|---|---|---|---|---|---|---|---|---|---|---|---|---|
| | LCA | ELCA | Top1 | LCA | ELCA | Top1 | LCA | ELCA | Top1 | LCA | ELCA | Top1 | LCA | ELCA | Top1 | LCA | ELCA | Top1 |
| ResNet18 He et al. (2016) | 6.643 | 7.505 | 0.698 | 6.918 | 7.912 | 0.573 | 8.005 | 9.283 | 0.202 | 8.775 | 8.853 | 0.330 | 8.449 | 9.622 | 0.011 | 8.062 | 8.636 | 0.272 |
| ResNet50 He et al. (2016) | 6.539 | **7.012** | **0.733** | 6.863 | **7.532** | **0.610** | 7.902 | 9.147 | 0.235 | 8.779 | **8.668** | 0.361 | 8.424 | **9.589** | 0.018 | 8.029 | **8.402** | 0.316 |
| CLIP_RN50 Radford et al. (2021) | 6.327 | **9.375** | 0.579 | 6.538 | **9.442** | 0.511 | 6.775 | 9.541 | 0.332 | 7.764 | 9.127 | 0.562 | 7.861 | 9.526 | 0.218 | 7.822 | 8.655 | 0.398 |
| CLIP_RN50x4 ?radford2021learning} | **6.166** | 9.473 | 0.641 | **6.383** | 9.525 | 0.573 | **6.407** | 9.518 | 0.415 | **7.435** | 8.982 | 0.681 | **7.496** | 9.388 | 0.384 | **7.729** | 8.354 | 0.504 |

Table 1: **Model performance corresponds to mistake severity. LCA ↓ / ELCA ↓ /Top1 ↑** indicate measurement on given dataset. We present two pairs of model comparisons from the VM and VLM families with different generalization abilities. We observe that models with higher Top 1 accuracy on OOD datasets typically have lower LCA and ELCA distances on OOD (except for ImageNet-v2, which is visually closer to ImageNet). Note that ELCA should not be compared across modalities, as it is sensitive to logit temperature.

The proposed ELCAD provides a more generalized metric for assessing model performance compared to Top 1, LCA distance and cross entropy. Top 1 accuracy only considers the top-ranked class; LCA distance measures the Top n class rankings but often treats each class equally (Bertinetto et al., 2020); Cross-entropy solely focuses on the model's assigned probability on the ground truth class, and

ELCA extends it to all classes. ELCAD captures the probabilistic distribution of mistake severity across all potential classes.

In Table 1, we empirically demonstrate that models with better OOD generalization (OOD Top 1 accuracy) also have lower LCAD/ELCAD.

# 3 THE SUITABILITY OF LCA AS A BENCHMARK FOR MODEL GENERALIZATION

This section explores the hypothesis that links taxonomy loss (LCA) with a model's generalization ability. Furthermore, we discuss how such insightful observations can be put into meaningful, actionable use.

**Obstacles to Model Generalization.** In traditional learning, models establish connections between image features and class labels. Nonetheless, such associations are subject to spurious correlations that may arise in the training data (Zhang et al., 2021). An example of this is erroneously associating the class 'ostriches' with the feature 'grass in the background' since 'ostriches' often appear in grasslands. These correlations are likely to fail when applied to an OOD dataset (Zhang et al., 2021).

**Essentials for Model Generalization.** Figure 3 demonstrates an OOD dataset, ImageNet-R, where, despite severe distribution shifts, humans can effortlessly identify the correct classes. This is because humans can identify the universally transferable semantic distinctions between classes as distinguishable feature for classification. Therefore, we posit that a model's generalization capabilities depends on the transferability of these learned features during training, and only semantic features that align with human understanding of object definitions are universally transferable to any OOD dataset.

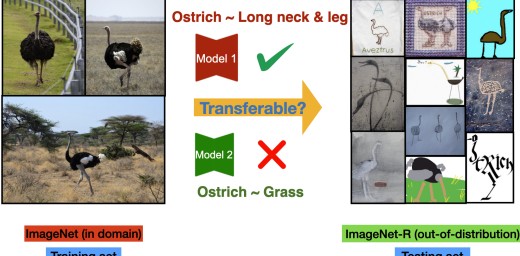

Figure 3: **Capturing transferable feature for model generalization**. Despite pronounced distribution shifts, ImageNet-R serves as a valid OOD test set for ImageNet classes as the images of ostriches, for instance, still maintain shape information (Geirhos et al., 2018) like "long neck", "big belly", and "long legs". We hypothesize that models exhibiting good generalization should capture these transferable semantic features rather than suffer from spurious correlation on feature like 'grass'.

But how can we measure what feature a model has learned during training? The decision-making process of deep neural networks trained end-to-end has become less interpretable. There have been attempts to decipher the decision process of models and form a decision-tree-like model (Wan et al., 2020; Gare et al., 2022), but these efforts have not linked this to an understanding of model generalization.

**Alignment to Class Taxonomy as representation measurement.**

Ideally, a model that captures more generalizable features tends to 'make better mistakes' by predict classes that are semantically closer to the ground truth class. As illustrate in Fig 4, model that learns to associate ostriches with features like 'long legs' and 'long neck', which are more transferable to OOD datasets, will likely predict classes like flamingos or Crane. In contrast, a model influenced by spurious correlations by falsely associate ostrich with grass, might predict a semantically distant class, like an Jaguars or Lions, which are also appear often on grass.

Our method involves measuring a model's generalization based on its in-domain semantic severity of mistake. We use the 'Least Common Ancestor' (LCA) distance, which is the taxonomic distance between the model's prediction and the ground truth class in a predefined taxonomy hierarchy, like WordNet. If a model consistently makes better mistakes on in-domain data, we can reasonably assume that the model has captured more transferable features for class discrimination.

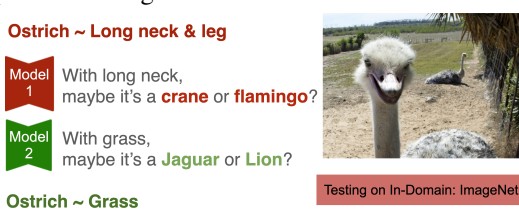

Figure 4: We hypothesize that models captured more transferable feature tend to predict classes that's semantically closer to ground truth.

**Class Taxonomy and Mistake Severity**: Class taxonomy or ontology has been widely utilized in the literature to indicate class formation (Deng et al., 2009a; Van Horn et al., 2018) and semantic relationships (Frome et al., 2013; Barz & Denzler, 2019; Wan et al., 2020; Redmon & Farhadi, 2017; Lin et al., 2022) between classes, offering a hierarchical organization of classes or categories. Following these works, we consider the WordNet class taxonomy (Miller, 1995) as an approximation of natural class taxonomy.

The severity of a mistake in many studies is quantified as the shortest path from the prediction node to the least common ancestor (LCA) in a predefined class hierarchy. This metric, known as 'LCA distance' or 'hierarchical error', was used in the early years of the ImageNet (Deng et al., 2009a) challenge. However, it was largely dismissed as it was widely believed to follow the same ordering as Top 1 accuracy (Deng et al., 2009a; Bertinetto et al., 2020). In this work, we revisit this metric and empirically demonstrate that Top 1 accuracy and LCA distance do not always align when VLMs are involved, which challenge the common notion. We also appeal for community attention to revisit this benchmark with its potential usage in measuring model's semantic awareness to indicate generalization.

**Causal/Invariant Representation Learning for OOD generalization.** Recently, there has been a notable increase in the field of OOD generalization research towards formulating training and testing distributions with causal structures (Arjovsky et al., 2019; Bühlmann, 2020; Peters et al., 2016), where the shifts in distribution primarily arise from interventions or confounding factors. Building upon this motivation, a series of methods have been proposed (Yang et al., 2021; Schölkopf et al., 2021; Shen et al., 2022; Subramanian et al., 2022) with the objective of achieving causal representation learning. For instance, CausalVAE (Yang et al., 2021). These methods leverage learned causal representations to capture the causal relationships underlying the data generation process (Kaur et al., 2022), which helps to mitigate the distributional shifts caused by interventions.

While the connection between OOD generalization and the causal concept is not entirely novel, those attempts have solely focused on the causal structure at the latent or abstract level, lacking both interpretability and transparency. Our method aligns with this growing interest in Causal/Invariant learning, which aims to capture the invariant latent data generation process. One should expect a model prediction that better align to the data generation process could be more robust under intervention thus generalize better. Although it's less feasible to model the data generation process of natural image (ImageNet), we essentially follow the same intuition and hypothesize that the WordNet class hierarchy serves as an approximation of the invariant relationship between class concepts. WordNet is a widely recognized and effective means of encoding semantic relationships between concepts, making it an appropriate proxy for aligning human semantic knowledge (Miller et al., 1990). Unlike previous work, the WordNet hierarchy provides interpretability, which adds a level of transparency to our understanding of model generalization.

## 4 EXPERIMENT

In this section, we are going to present experiment benchmarking relationship between LCA and generalization.

**Setup** This paper leverages 75 pretrained models sourced from open repositories on GitHub for empirical analysis. Our selection includes 36 Vision Models (VMs) pretrained on ImageNet, and 39 Vision-Language Models (VLMs), which incorporate language as part of the supervision. A comprehensive list of the model details will be provided in C to ensure reproducibility. In this work, we use *ImageNet*(Deng et al., 2009a) as the source in-distribution (ID) dataset, while *ImageNet-v2*(Recht et al., 2019), *ImageNet-Sketch*(Hendrycks & Dietterich, 2019), *ImageNet-Rendition*(Hendrycks et al., 2021), *ImageNet-Adversial*(Hendrycks et al., 2021), and *ObjectNets*(Barbu et al., 2019) are adopted as out-of-distribution datasets, exemplifying natural distribution shift. We utilize the ImageNet hierarchy as depicted in (Bertinetto et al., 2020).

For our correlation experiment, we employ $R^2$ *(Coefficient of Determination)* and *PEA (Pearson correlation coefficient)* to measure the strength and direction of linear relationships between two variables. In addition, we use *KEN (Kendall rank correlation coefficient)* and *SPE (Spearman rank-order correlation coefficient)* to assess the correspondence of the rankings of two variables.

The importance of these measurements lies in their different focus. Linearity measures, such as $R^2$ and PEA, are primarily interested in the fit of a linear model to the data points, allowing us to quantify the predictability of the changes in one variable based on the other. Ranking measures, like KEN and SPE, on the other hand, provide insights into how the rankings of the variables relate to each other, which is particularly vital in downstream applications such as image retrievals and search engine optimization, where understanding and predicting the ordering of data points is often more important than predicting their exact values. For prediction experiments, we utilize MAE (Mean Absolute Error) to quantify the absolute difference between prediction and ground truth.

Although *ImageNet-v2* is predominantly deemed an OOD dataset in most prior literature (Shankar et al., 2020; Miller et al., 2021; Baek et al., 2022), our experiments suggest that *ImageNet-v2* aligns more closely with ImageNet than with other OOD datasets; we delve into these details in D.

| | Element | | ImageNetv2 | | ImageNet-S | | ImageNet-R | | ImageNet-A | | ObjectNet | |
|---|---|---|---|---|---|---|---|---|---|---|---|---|
| | ID | OOD | R^2 | PEA | R^2 | PEA | R^2 | PEA | R^2 | PEA | R^2 | PEA |
| ALL | Top1 | Top1 | **0.962** | **0.980** | 0.075 | 0.275 | 0.020 | 0.140 | 0.009 | 0.094 | 0.273 | 0.522 |
| | LCA | Top1 | 0.339 | 0.582 | **0.838** | **0.915** | **0.779** | **0.883** | **0.869** | **0.932** | **0.915** | **0.956** |
| | Top1 | Top5 | **0.889** | **0.943** | 0.052 | 0.229 | 0.004 | 0.060 | 0.013 | 0.115 | 0.262 | 0.512 |
| | LCA | Top5 | 0.445 | 0.667 | **0.883** | **0.940** | **0.738** | **0.859** | **0.909** | **0.953** | **0.924** | **0.961** |
| VLM | Top1 | Top1 | **0.996** | **0.998** | 0.860 | 0.927 | 0.851 | 0.923 | 0.578 | 0.761 | **0.945** | **0.972** |
| | LCA | Top1 | 0.956 | 0.978 | **0.922** | **0.960** | **0.889** | **0.943** | **0.792** | **0.900** | 0.936 | 0.968 |
| | Top1 | Top5 | **0.988** | **0.994** | 0.867 | 0.931 | 0.820 | 0.906 | 0.740 | 0.860 | **0.970** | **0.985** |
| | LCA | Top5 | 0.930 | 0.964 | **0.949** | **0.974** | **0.848** | **0.921** | **0.828** | **0.910** | 0.931 | 0.965 |
| VM | Top1 | Top1 | **0.996** | **0.998** | 0.824 | 0.908 | **0.801** | **0.895** | 0.523 | 0.723 | 0.900 | 0.949 |
| | LCA | Top1 | 0.976 | 0.988 | **0.895** | **0.945** | 0.768 | 0.877 | **0.833** | **0.912** | 0.913 | 0.956 |
| | Top1 | Top5 | **0.993** | **0.997** | 0.829 | 0.910 | **0.821** | **0.906** | 0.696 | 0.834 | 0.919 | 0.959 |
| | LCA | Top5 | 0.970 | 0.985 | **0.925** | **0.962** | 0.777 | 0.882 | **0.925** | **0.962** | 0.936 | 0.967 |

Table 2: **Correlation measurement of ID LCA/Top1 with OOD Top1/Top5** on 75 models across modality (36 VMs and 39 VLMs) following Fig 5. The 'ALL grouping' demonstrates that LCA has a strong correlation with OOD performance on all datasets (except ImageNet-v2). We take the absolute value of all correlations for simplicity. Equivalently, LCA is also a very good OOD indicator when only involved VM or VLM.

## 4.1 LCA-ON-THE-LINE: IN-DOMAIN TAXONOMY DISTANCE (LCA) AS AN OUT OF DISTRIBUTION (OOD) PERFORMANCE BENCHMARK

The model's in-distribution (ID) accuracy and its out-of-distribution (OOD) accuracy are largely considered to be strongly correlated, as corroborated by (Miller et al., 2021). This potent correlation forms a significant baseline for comparison in our research. Differing from the framework presented in (Miller et al., 2021) that only compare models within same modality, our work fill in the gap to contrast model of different modality, involving Vision Models (VM) trained on ImageNet, and Vision-Language Models (VLM) trained on Laion. In addition to the Top1 OOD accuracy, we incorporated Top5 OOD accuracy, yielding a more holistic evaluation of model generalization.

As displayed in table 2, the ImageNet in-domain accuracy (Miller et al., 2021) forms a robust predictor for most OOD datasets when the comparison is limited to models with similar setups (VM or VLM). However, this predictor falls short when attempting to unify models of different modality. As highlighted in Fig 5 (indicated in red), when adhering to 'accuracy on the line' (Miller et al., 2021), all four OOD datasets plotted showcase two distinct linear trends, representing models that belong to the VM and VLM families. This observation aligns with (Cherti et al., 2022), where it was found that VLM models, despite exhibiting significantly lower ID accuracy, could attain higher OOD performance than their state-of-the-art VM counterparts. As a consequence, in-domain accuracy (Miller et al., 2021) fail to explain this misalignment between generalization of VMs and VLMs.

As shown in Fig 6, our method adopting in-domain LCA score could restore the linear trends among model of different modality. As demonstrated in table 2 and Fig 5 (colored in green), the severity of in-domain errors serves as a more effective indicator of model performance compared to in-domain accuracy. It consistently exhibits a strong linear correlation with all OOD benchmark accuracies for natural distribution shifts (both $R^2$ and the Pearson correlation coefficient approach 0.9).

Notably, our experiments showed that (Miller et al., 2021) is a more reliable indicator solely for ImageNet-v2, given its visual similarity to ImageNet. Please refer to D for discussion. In G, we will

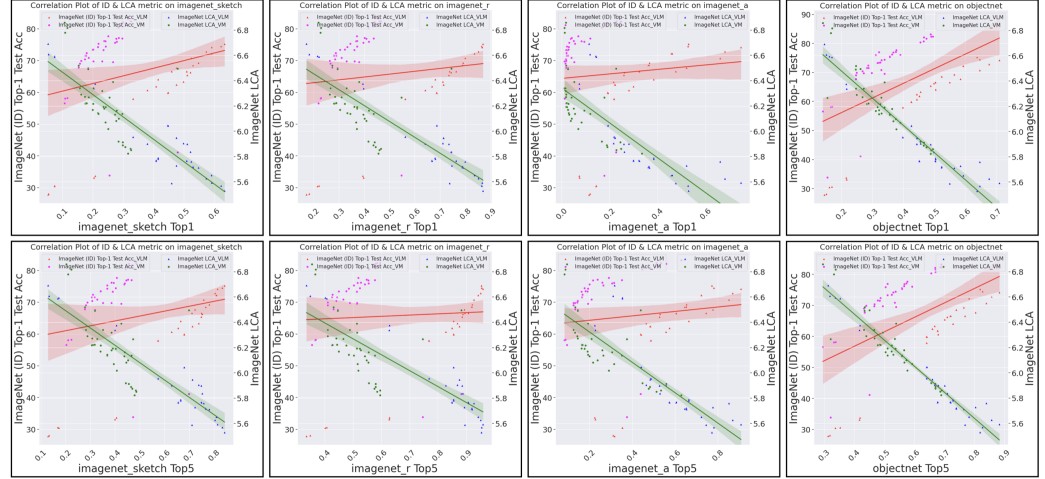

Figure 5: **Correlating OOD Top-1/Top-5 Accuracy (VM+VLM, 75 models) on 4 ImageNet-OOD Datasets.** Following Tab 2. Each plot's x-axis represents the OOD dataset metric (with OOD Top-1 in the top row, and OOD Top-5 accuracy in the bottom row); Red represents in-domain classification accuracy (Top-1); Green denotes in-domain taxonomy distance (LCA). The plots clearly demonstrate that the in-domain LCA has a strong correlation with the model's OOD performance across all OOD datasets. Even though in-domain Top-1 accuracy is widely considered a good OOD performance indicator (Miller et al., 2021), it falls short in providing a unified metric encompassing both VMs and VLMs. As seen, the plots often exhibit a pattern of two distinct lines rather than a single line. If necessary, please find png of this image in supplementary for better legibility.

also include measurements from the KEN and SPE, which similarly demonstrate robust scores in preserving the relative ordering of model OOD performance.

## 4.2 PREDICTING OOD PERFORMANCE WITH IN DOMAIN LCA

We further highlight the effectiveness of the 'LCA-on-the-Line' approach by estimating model OOD performance using a linear function derived from in-domain LCA scores. For comparison, we included four competitive baselines: Average Confidence (AC), which leverages the OOD logit after temperature scaling; two methods from Agreement-on-the-Line (Aline-D and Aline-S), which utilize consensus of pairs of models on OOD benchmarks; and 'Accuracy on the Line' (ID Top1), which employs the in-domain accuracy of established measurement models to fit a linear function. Furthermore, instead of performing a probit transform as done in (Baek et al., 2022) and (Miller et al., 2021), we implemented min-max scaling because LCA does not fall within the [0,1] range.

As illustrated in Table 3, in-domain LCA proves to be a significantly more robust OOD error predictor than other baselines across four OOD

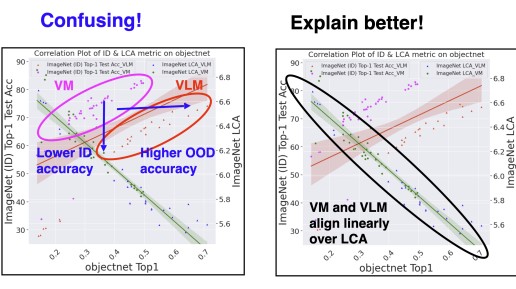

Figure 6: Our method restore the "on-the-line" linear relationship by unifying both VMs and VLMs. Our method provide a compelling alternative to understand why vision-language models with lower in-domain accuracy might generalize better to OOD datasets than vision models.

benchmarks with varying distribution shifts. This robustness is especially apparent for ImageNet-A, an adversarial dataset derived from the misclassification of ResNet50 on ImageNet. Consequently, models pre-trained on ImageNet tend to underperform on this dataset, particularly those with lower accuracy than ResNet50. This leads to a decrease in robustness for the in-domain accuracy(Miller et al., 2021), methods calibrated from in-domain validation sets (Hendrycks & Gimpel, 2017), and OOD agreement of models from different families (Baek et al., 2022). In contrast, the LCA, which

relies solely on the relative ranking of class predictions from a single model, is less sensitive to these issues and thus delivers more consistent performance. This further underscores the efficacy of the LCA as a powerful predictor in challenging OOD scenarios.

### 4.3 ENHANCING GENERALIZATION THROUGH CLASS TAXONOMY ALIGNMENT.

Building upon the earlier discussion, we explore how the devised benchmarking method can be utilized to enhance a model's generalization capability.

**Inferring Class Taxonomy from a Pretrained Model Using K-Means Clustering**. While the number of publicly available datasets providing class taxonomy is limited (Deng et al., 2009a; Van Horn et al., 2018), the usefulness of such taxonomy is unquestionable. Hence, we propose a method to construct a latent class taxonomy, expanding the potential applications of our work.

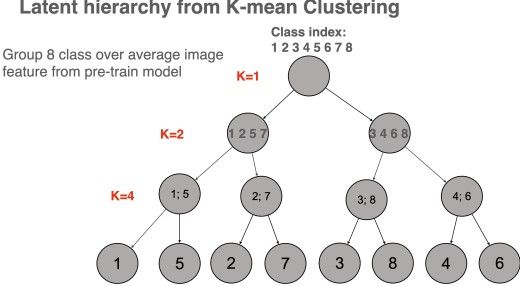

The essence of class taxonomy lies in its representation of the inter-class distance, encoding class proximity and identifying which classes cluster closely in semantic space. In this spirit, we construct a class taxonomy matrix using K-means clustering. Experiment in Tab 4 shows that our method is very robust regardless which model were used to construct class hierarchy. As illustrated in Fig 7, we adopt average class features to cluster data hierarchically at 10 different levels, with increasing number of cluster to indicate class adjacency. Implementation detail in F.2 in appendix.

Figure 7: **Visualization of K-mean clustering process over 8 class.**

|  |  | ImageNetv2 | ImageNet-S | ImageNet-R | ImageNet-A | ObjectNet |
|---|---|---|---|---|---|---|
| ALL | ID Top1 (Miller et al., 2021) | **0.040** | 0.230 | 0.277 | 0.192 | 0.178 |
|  | AC (Hendrycks & Gimpel, 2017) | 0.043 | 0.124 | 0.113 | 0.324 | 0.127 |
|  | Aline-D (Baek et al., 2022) | 0.121 | 0.270 | 0.167 | 0.409 | 0.265 |
|  | Aline-S (Baek et al., 2022) | 0.072 | 0.143 | 0.201 | 0.165 | 0.131 |
|  | (Ours) ID LCA | 0.162 | **0.078** | **0.107** | **0.061** | **0.048** |
| VLM | ID Top1 (Miller et al., 2021) | **0.014** | 0.077 | 0.064 | 0.127 | 0.052 |
|  | AC (Hendrycks & Gimpel, 2017) | 0.029 | **0.050** | **0.044** | 0.217 | 0.088 |
|  | Aline-D (Baek et al., 2022) | 0.151 | 0.250 | 0.081 | 0.296 | 0.260 |
|  | Aline-S (Baek et al., 2022) | 0.070 | 0.069 | 0.068 | **0.080** | 0.153 |
|  | (Ours) ID LCA | 0.047 | 0.059 | 0.062 | 0.094 | **0.043** |
| VM | ID Top1 (Miller et al., 2021) | **0.013** | 0.099 | 0.108 | 0.143 | 0.068 |
|  | AC (Hendrycks & Gimpel, 2017) | 0.059 | 0.204 | 0.188 | 0.441 | 0.168 |
|  | Aline-D (Baek et al., 2022) | 0.083 | 0.427 | 0.313 | 0.665 | 0.364 |
|  | Aline-S (Baek et al., 2022) | 0.105 | 0.182 | 0.092 | 0.574 | 0.216 |
|  | (Ours) ID LCA | 0.029 | **0.079** | **0.113** | **0.080** | **0.056** |

Table 3: **Error Prediction of OOD Datasets** across 75 models of diverse settings with **MAE loss ↓**. Top1 in **bold** and Top2 in underline. Despite ImageNet's in-domain accuracy maintain as a significant indicator of ImageNet-v2 accuracy, the in-domain LCA outperforms it as a robust error predictor across four naturally distributed OOD datasets, particularly ImageNet-A, which stumps other methods.

| 75 LCA Stats | Element | | ImageNetV2 | ImageNet-S | ImageNet-R | ImageNet-A | ObjectNet |
|---|---|---|---|---|---|---|---|
|  | ID | OOD |  |  |  |  |  |
| Top1_corr | Top1 | Top1 | **0.980** | 0.274 | 0.141 | 0.093 | 0.522 |
| LCA_corr (Mean) | LCA | Top1 | 0.815 | **0.773** | **0.712** | **0.662** | **0.930** |
| LCA_corr (Min) | LCA | Top1 | 0.721 | 0.715 | 0.646 | 0.577 | 0.890 |
| LCA_corr (Max) | LCA | Top1 | 0.863 | 0.829 | 0.780 | 0.717 | 0.952 |
| LCA_corr (std) | LCA | Top1 | 0.028 | 0.022 | 0.027 | 0.025 | 0.010 |

Table 4: **Correlation Measurement between ID LCA/Top1 and OOD Top1 across 75 Latent Hierarchies Derived from K-means**. For each pretrained model, we constructed a 75-class taxonomy hierarchy using the K-means clustering method described previously. We then calculated the LCA for each hierarchy as an in-domain indicator and compared it to the OOD accuracy using the same settings as in 2. This shows our hierarchy construction method is robust across all pretrained models.

**Employing Class Taxonomy as Soft Labels**. We propose a straightforward approach to demonstrate the potential of LCA as a benchmarking tool for generalization. We encode the normalized pairwise

LCA between each class as soft labels and apply linear probing over the pretrained model. Contrary to the rigid probabilistic distribution of single-label classification, we formulate the problem as multi-labeling. We employ a sigmoid-style (Beyer et al., 2020) BCE loss instead of softmax, relaxing the constraints on inter-class interaction. A more detailed setup will be included in the appendix.

Following method above, we have constructed class taxonomy matrices for AlexNet (Krizhevsky et al., 2017) and Swin Transformer (Liu et al., 2021), which respectively represent the best and worst performing models on ImageNet in our model pool. Intriguingly, the hierarchy constructed from the model's pretrained features partially encapsulates the model's interpretation of interclass relationships. As table 5 illustrates, incorporating accurate inter-class distance consistently enhances OOD performance across all four OOD benchmarks, albeit with a slightly lower Top 1 accuracy.

However, this approach does lead to a slight drop in in-domain accuracy as it less intensively optimizes towards the ground truth class. Inspired by the notion that models are more confident where they excel (Wortsman et al., 2022), we apply linear interpolation between linear layers trained from cross-entropy and our proposed loss function. The results suggest that this method strikes a balance, delivering competitive performance on both ID and OOD datasets.

Importantly, we find that models using hierarchies constructed from pretrained models fall short in OOD generalization compared to those utilizing WordNet hierarchy, even though they exhibit slightly improved ID performance. This indicates that enforcing arbitrary inter-class relationships, derived from in-domain datasets, can negatively affect OOD performance.

For result of using class taxonomy base prompt engineering on zero-shot vision-language models, please **refer to appendix**.

|  | ImageNet | ImageNetv2 | ImageNet-S | ImageNet-R | ImageNet-A | ObjectNet |
|---|---|---|---|---|---|---|
| Baseline | **0.690** | **0.5618** | **0.199** | 0.322 | 0.010 | 0.267 |
| AlexNet Hier | 0.665 | 0.5402 | 0.189 | 0.294 | 0.017 | 0.247 |
| Swin-T Hier | 0.668 | 0.5429 | 0.196 | 0.312 | 0.023 | 0.259 |
| WordNet Hier | 0.664 | 0.5387 | **0.199** | 0.329 | **0.024** | **0.272** |
| (CE + CE) Interp | **0.695** | 0.5645 | 0.196 | 0.325 | 0.011 | 0.273 |
| (AlexNet + CE) Interp | 0.694 | 0.5665 | 0.200 | 0.325 | 0.012 | 0.274 |
| (Swin-T + CE) Interp | **0.695** | **0.5694** | 0.202 | 0.331 | 0.012 | 0.274 |
| (WordNet + CE) Interp | 0.694 | 0.5638 | **0.2073** | **0.335** | **0.014** | **0.282** |

Table 5: **Interpolating Class Taxonomy to Linear Probing on ResNet18 Feature**. The top table displays results from models trained using a class hierarchy constructed from the indicated model via K-means. The bottom table presents the results of the aforementioned models when interpolated with layers trained from cross entropy in the weight space (Wortsman et al., 2022). Training with a WordNet hierarchy delivers the most significant improvements across OOD benchmarks despite lower Top 1 accuracy, whereas models using hierarchies inferred from pretrained models yield lesser gains.

## 5 LIMITATIONS, CONCLUSIONS, AND FUTURE DIRECTIONS

While we benchmarked and used LCA based on class hierarchy to measure model generalization, the findings from this work indicate that it is not an effective indicator for datasets visually similar to In-domain data (like ImageNet2). For these datasets, In-domain Top1 remains a strong indicator, which potentially limits the utility of LCA. Also, it's expected that LCA will shows a weaker discrimination between models on datasets with small number of class (like Cifar (Krizhevsky et al.)).

In conclusion, this work reinvigorates the LCA distance using WordNet hierarchy as a benchmark for model OOD generalization. WordNet's class taxonomy represents a form of semantic knowledge that aligns with human cognition of class relationships. Ideally, models that capture correct semantic representation should make fewer severe mistakes. We discovered that severity of in domain mistakes (i.e., the ability to capture WordNet ontology) has strong relationship with model's OOD Top 1 accuracy across multiple ImageNet-OOD datasets. This relationship is not reflected when using the widely-accepted in-domain Top 1 accuracy (Miller et al., 2021) as a measurement when comparing vision-only and vision-language zero-shot models. Furthermore, we demonstrated that aligning model predictions with class taxonomy, whether through prompt engineer or introducing regularization loss, can enhance model generalization. Future direction could focus on provide theoretical justification under LCA-on-the-line, and perform larger scale empirical study regarding this benchmark. This work provides new insights into model generalization using existing resources and encourages further investigation in this direction.

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

## A    REPRODUCIBILITY STATEMENT

In an effort to promote transparency and ease of replication, our code will be made publicly available upon acceptance of this paper. Detailed hyperparameters and experimental procedures will be comprehensively documented in the appendix. All experiments were conducted under fixed random seed conditions to ensure consistent outputs. Furthermore, we utilized checkpoints retrieved from publicly accessible codebases. These steps have been taken to provide a solid foundation for replication and extension of our work. Hence, the results presented in this paper should be easily reproducible, fostering further research in this domain.

## B    ETHICS STATEMENT

While this research primarily serves to deepen our understanding of model generalization mechanisms, it is imperative to acknowledge the potential for misuse. The methods proposed in this work could conceivably be leveraged to guide adversarial attacks aimed at reducing the generalization capabilities of existing models. While our research was not specifically intended for such purposes, it is crucial to be cognizant of the duality that our understanding of model generalization could bring about. This potential for exploitation underscores the importance of developing robust, secure models, and implementing ethical guidelines for the deployment of such knowledge.

## C    MODEL ARCHITECTURES

We list all models used in ours experiment as follows, including 36 Vision Only Models ( VM ) and 39 Vision-Language Models ( VLM ).

| Model Category | Architecture | Number of models | Checkpoint Link |
|---|---|---|---|
| VM (Vision-Only-Models) | AlexNet (Krizhevsky et al., 2017) | 1 | alexnet |
| | ConvNeXt (Liu et al., 2022) | 1 | convnext$_{tiny}$ |
| | DenseNet (Huang et al., 2017) | 4 | densenet121
densenet161
densenet169
densenet201 |
| | EfficientNet (Tan & Le, 2019) | 1 | efficientnet_b0 |
| | GoogLeNett (Szegedy et al., 2015) | 1 | googlenet |
| | Inceptionv3 (Szegedy et al., 2016) | 1 | inceptionV3 |
| | MnasNet (Tan et al., 2019) | 4 | mnasnet0.5
mnasnet0.75
mnasnet1.0
mnasnet1.3 |
| | Mobilenet-V3 (Howard et al., 2019) | 2 | mobilenetv3_small
mobilenetv3_large |
| | Regnet (Radosavovic et al., 2020) | 1 | regnet_y_1_6gf |
| | Wide ResNet (Zagoruyko & Komodakis, 2016) | 1 | wide_resnet101_2 |
| | ResNet (He et al., 2016) | 5 | resnet18
resnet34
resnet50
resnet101
resnet152 |
| | ShuffleNet (Zhang et al., 2018) | 1 | shufflenet_v2_x2_0 |
| | SqueezeNet (Iandola et al., 2016) | 2 | squeezenet1_0
squeezenet1_1 |
| | Swin Transformer (Liu et al., 2021) | 1 | swin_b |
| | VGG (Simonyan & Zisserman, 2015) | 8 | vgg11
vgg13
vgg16
vgg19
vgg11_bn
vgg13_bn
vgg16_bn
vgg19_bn |
| | ViT (Dosovitskiy et al., 2020) | 2 | vit_b_32
vit_l_32 |
| VLM (Vision-Language-Models) | ALBEF (Li et al., 2021) | 1 | albef_feature_extractor |
| | BLIP (Li et al., 2022b) | 1 | blip_feature_extractor_base |
| | CLIP (Radford et al., 2021) | 7 | RN50
RN101
RN50x4
ViT-B-32.pt
ViT-B-16.pt
ViT-L-14.pt
ViT-L-14-336px |
| | OpenCLIP (Cherti et al., 2023) | 30 | openCLIP:
openCLIP_('RN101', 'openai')
openCLIP_('RN101', 'yfcc15m')
openCLIP_('RN101-quickgelu', 'openai')
openCLIP_('RN101-quickgelu', 'yfcc15m')
openCLIP_('RN50', 'cc12m')
openCLIP_('RN50', 'openai')
openCLIP_('RN50', 'yfcc15m')
openCLIP_('RN50-quickgelu', 'cc12m')
openCLIP_('RN50-quickgelu', 'openai')
openCLIP_('RN50-quickgelu', 'yfcc15m')
openCLIP_('RN50x16', 'openai')
openCLIP_('RN50x4', 'openai')
openCLIP_('RN50x64', 'openai')
openCLIP_('ViT-B-16', 'laion2b_s34b_b88k')
openCLIP_('ViT-B-16', 'laion400m_e31')
openCLIP_('ViT-B-16', 'laion400m_e32')
openCLIP_('ViT-B-16-plus-240', 'laion400m_e31')
openCLIP_('ViT-B-16-plus-240', 'laion400m_e32')
openCLIP_('ViT-B-32', 'laion2b_e16')
openCLIP_('ViT-B-32', 'laion2b_s34b_b79k')
openCLIP_('ViT-B-32', 'laion400m_e31')
openCLIP_('ViT-B-32', 'laion400m_e32')
openCLIP_('ViT-B-32', 'openai')
openCLIP_('ViT-B-32-quickgelu', 'laion400m_e31')
openCLIP_('ViT-B-32-quickgelu', 'laion400m_e32')
openCLIP_('ViT-L-14', 'laion2b_s32b_b82k')
openCLIP_('ViT-L-14', 'laion400m_e31')
openCLIP_('ViT-L-14', 'laion400m_e32')
openCLIP_('coca_ViT-B-32', 'laion2b_s13b_b90k')
openCLIP_('coca_ViT-L-14', 'laion2b_s13b_b90k') |

## D DISCUSSION

**Reestablishing LCA as a Comprehensive Measure of Model Generalization.** While Top 1 ID accuracy shows a pronounced linear trend with OOD datasets when models follow similar training mechanisms, the relationship blurs with vision-only and VLMs — a phenomenon observed in early work (Fang et al., 2022; Wortsman et al., 2022; Cherti et al., 2022). This correlation could elucidate the unexpected outcome where zero-shot VLMs with lower top-1 accuracy outperform competitive vision models when generalizing to unfamiliar datasets. While several works suggest that the data diversitysignificantly impacts generalization (Fang et al., 2022; Schuhmann et al., 2022; Kaur et al., 2022), our results imply that the LCA could offer a more holistic evaluation of model generalization. By taking into account elements such as training data size, architecture, loss, and more, LCA allows for a more complete measure of model ability to capture correct semantic distinctions shared across ID and all OOD benchmarks. This establishes a comprehensive benchmark that encapsulates various generalization factors and mitigates the inflation of VLM on "Effective Robustness" (Taori et al., 2020). We encourage future work to conduct large-scale analytic studies on generalization factors in tandem with the LCA.

**Is it Possible for a Semantically-Aware (Low LCA) Model to Have Low Top 1 Accuracy?** Our empirical analyses reveal a correlation: models in the wild (not deliberately tuned on class taxonomy) with lower Top 1 accuracy tend to have higher LCA distances. However, this relationship is correlative rather than causal. It remains possible to adversarially design a model that consistently predicts the semantically closest class to the true class, where the model would exhibit a low LCA distance while maintaining zero Top 1 accuracy. Thus, while a correlation exists between Top 1 and LCA, causality cannot be implied, and this relationship can be disrupted under intentional adversarial training.

**Does ImageNet LCA (Taxonomy Distance) Reflect ImageNet Top 1 Accuracy?** Literature often posits that LCA and Top-1 accuracy follow the same trend on same dataset (Deng et al., 2009a; Bertinetto et al., 2020). Intuitively, a high perform model would better fit data distribution, leads to fewer severe errors. This trend generally holds true when considering only models under similar settings (either VM or VLM). However, when including both VM and VLM models, ImageNet and ImageNet-v2 show a weak correlation between LCA and Top-1 accuracy, while other semantically distinct OOD datasets exhibit a stronger relationship. This challenges the prevailing belief that in domain Top-1 accuracy and LCA maintain same ranking (Deng et al., 2009b; Bertinetto et al., 2020).

**ImageNet-v2 Demonstrates Similar Class Discrimination Features to ImageNet.** ImageNet-v2, a recollection of the ImageNet, is frequently used as an OOD dataset for ImageNet in various studies (Shankar et al., 2020; Miller et al., 2021; Baek et al., 2022). Nonetheless, as shown in table 2 above and Figure 4 and Table 3 in appendix, our experiments suggest that ImageNet-v2 bears more resemblance to ImageNet than other OOD datasets. We hypothesize that fewer external interventions in ImageNet-v2's data generation process lead to visual similarity to ImageNet, allows even spurious relationships encoded from ImageNet to successfully transfer to ImageNet-v2. Thus model pretrained on imageNet (VMs) will inflate the accuracy on ImageNetv2, preventing it from aligning with trend from VLMs.

## E METRIC

In this section, we outline the metrics adopted for our experiment.

### E.1 CORRELATION MEASUREMENT

Correlation measurements quantify the degree of association between two variables. This can be further subdivided into linearity and ranking measurements.

#### E.1.1 LINEARITY MEASUREMENT

Linearity measurement evaluates the strength and direction of a linear relationship between two continuous variables. We use the $R^2$ and Pearson correlation coefficients to assess linearity.

**R² (Coefficient of determination)**: The R², or coefficient of determination, quantifies the proportion of the variance in the dependent variable that can be predicted from the independent variable(s). It ranges from 0 to 1, where 1 indicates perfect predictability. It is defined as:

$$R^2 = 1 - \frac{\sum_{i=1}^{n}(y_i - f(x_i))^2}{\sum_{i=1}^{n}(y_i - \bar{y})^2} \tag{1}$$

where $f(x_i)$ is the prediction of $y_i$ from the model, $\bar{y}$ is the mean of the actual $y$ values, and $n$ is the number of data points.

**PEA (Pearson correlation coefficient)**: The Pearson correlation coefficient, denoted as $r$, measures the linear relationship between two datasets. It is defined as:

$$r = \frac{\sum_{i=1}^{n}(x_i - \bar{x})(y_i - \bar{y})}{\sqrt{\sum_{i=1}^{n}(x_i - \bar{x})^2}\sqrt{\sum_{i=1}^{n}(y_i - \bar{y})^2}} \tag{2}$$

where $\bar{x}$ and $\bar{y}$ are the mean values of the datasets $x$ and $y$, respectively, and $n$ is the number of data points.

### E.1.2  RANKING MEASUREMENT

Ranking measurement evaluates the degree of correspondence between the rankings of two variables, even when their relationship is non-linear. The Kendall and Spearman rank correlation coefficients are metrics used for this purpose.

**KEN (Kendall rank correlation coefficient)**: Also known as Kendall's tau ($\tau$), this coefficient measures the ordinal association between two variables. It is defined as:

$$\tau = \frac{(\text{number of concordant pairs}) - (\text{number of discordant pairs})}{\frac{1}{2}n(n-1)} \tag{3}$$

where $n$ is the number of data points.

**SPE (Spearman rank-order correlation coefficient)**: The Spearman rank-order correlation coefficient, denoted as $\rho$, assesses the monotonic relationship between two variables. It is defined as:

$$\rho = 1 - \frac{6\sum_{i=1}^{n} d_i^2}{n(n^2-1)} \tag{4}$$

where $d_i$ is the difference between the ranks of corresponding data points in the two datasets and $n$ is the number of data points.

### E.2  TAXONOMY MEASUREMENT

Taxonomy measurement is designed to assess the alignment between the model-predicted class ranking and the predefined class taxonomy hierarchy tree. This is also referred to as 'mistake severity' or 'taxonomy distance'.

### E.2.1  LCA DISTANCE

Following (Bertinetto et al., 2020; Valmadre, 2022), we define LCA distance using a predefined hierarchy tree, as indicated in Fig2. We adopt class distance in a hierarchical tree format to denote inter-class relationships, which is necessary to calculate LCA and ELCA. Given a ground truth node y (node 1 in the plot) and a model prediction node $y'$ (node 3 in the plot), their LCA node $lca(y, y')$ is node 6 in the plot. We define it as:

$$lcad(y', y) := f(lca(y', y)) - f(y), \tag{5}$$

where $f(\cdot)$ represents a function for a node's score, such as the tree depth or information content.

**Scores as tree depths**: We define a function $d(x)$ to retrieve the depth of node x from tree T. Then, LCA distance is defined as:

$$lcad(y', y)_d := (d(y) - d(lca(y', y))) + (d(y') - d(lca(y', y))), \tag{6}$$

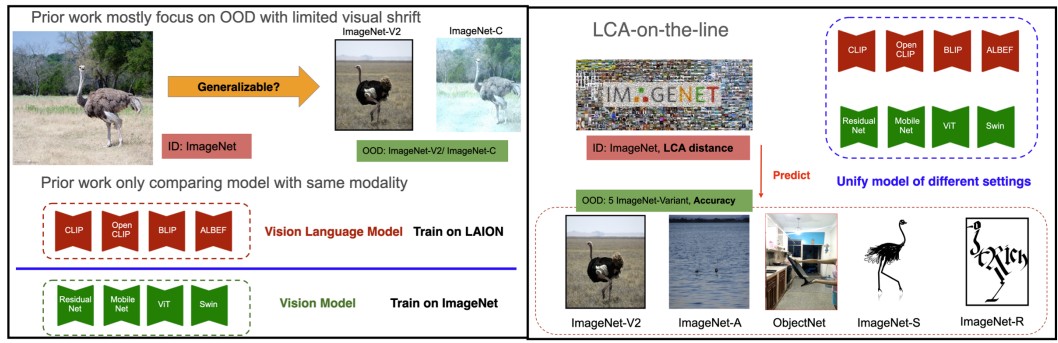

Figure 8: **Illustration of settings comparison to prior work.** Left: prior work settings; Right: our settings for LCA-on-the-Line

where we also append $d(lca(y', y)) - d(y')$ to counter tree imbalance.

**Scores as information**: Defining score as tree depth may be vulnerable to an imbalanced hierarchical tree. Thus, we also define a node's score as information to put more weight on nodes with more descendants. Formally, following (Valmadre, 2022), we apply a uniform distribution p to all leaf nodes in the tree that indicate a class in the classification task. The probability of each intermediate node in the tree is calculated by recursively summing the scores of its descendants. Then, the information of each node is calculated as $i(node) := -log2(p)$. The LCA distance is then defined as:

$$lcad_i(y', y) := i(y) - i(lca(y', y)), \tag{7}$$

In this work, we adopt $lcad_i(y', y)$ for objectNet, ImageNet-R, and ImageNet-v2, and $lcad_d(y', y)$ for ImageNet-S, and ImageNet-A to achieve optimal performance. Both metrics can significantly outperform Top1 in-domain accuracy.

### E.3 ELCA DISTANCE

We define ELCA as a more general form of LCA distance; it's a weighted combination of each leaf node [1,2,3,4] as in Fig 2, weighted by class probability. Formally, for each prediction node, the probabilistic distribution over all candidate classes can be obtained by applying a softmax function $softmax(x) : \mathbb{R} \to [0, 1]$ to get model outputs probability $(\widehat{p}i, 1, \ldots, \widehat{p}K, 1)$ over the $K$ classes (e.g., 1000 in ImageNet). The ELCA distance can then be defined as:

$$ELCAD(model, \mathcal{D}) := \frac{1}{nK} \sum_{i=1}^{n} \sum_{k=1}^{K} \widehat{p}_{k,i} \cdot lcad(k, y_i) \tag{8}$$

## F EXPERIMENT SETUP

### F.1 SETUP COMPARE TO PRIOR WORK

Fig 8 shows the setting comparision between prior work and our work. To the best of our knowledge, LCA-on-the-line is the first approach to uniformly measure model robustness across different model modalities and OOD datasets with significant distribution shifts.

### F.2 K-MEAN CLUSTERING FOR LATENT CLASS HIERARCHY CONSTRUCTION

As shown in Fig 7, we start with a pretrained model $M$, in-domain image data $X$, and labels $y$ for $k$ classes. We first extract the in-domain data features $M(X)$. Knowing the labels, we can categorize $M(X)$ by $y$, resulting in $k$ average class features, denoted as $KX$. Using these per-class features, we perform a 10-layer hierarchical clustering. For $KX$, we execute the K-means algorithm where

the number of cluster centers is $2^i$, with $i$ in the range of $1, 2, 3, 4, ...9$ as $2^9 < 1000$. This process results in 9 cluster outcomes. Subsequently, we compute the pairwise LCA between the $k$ classes, establishing the cluster level where both classes share the same cluster as the height of LCA. By definition, all classes have a base cluster level of 10.

### F.3 Loss for Linear Probing Experiment

For our linear probing experiment, we define our loss function as follows. For a class with n classes, we first define an n*n LCA distance matrix M, where M[i,k] indicates pairwise LCA distance $lcad(i, k)$, where lca is calculated from either using WordNet hiearchy, or hierarchy constructed from K-mean algorithm(introduced in the main paper). Then, we scale M by applying an exponential function, MinMax scaling, and normalize to 1 for each row, i.e., M = $normRow(minmaxScaling(M.exp()))$. In computing the loss, we use Binary Cross Entropy (BCE) and adopt the corresponding row value as a soft label. Specifically, if class-i is the ground truth for the given data, we use M[i,:] as the soft label.

### F.4 LCA matrix from pretrain model

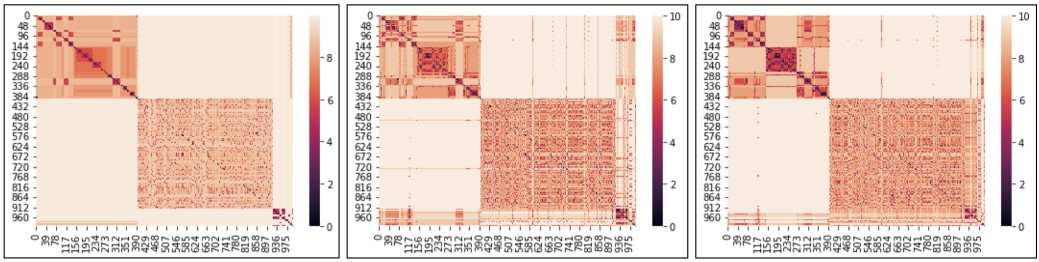

Figure 9: **Comparison between LCA distance matrices**. From left to right: WordNet hierarchy; matrix constructed from AlexNet (Krizhevsky et al., 2017); and matrix constructed from CLIP ResNet50 (Radford et al., 2021). We observe a higher alignment between the CLIP RN50 LCA distance matrix and the WordNet hierarchy as compared to the one from AlexNet.

We showcase an example of LCA distance matrix comparison in Figure 9, with the diagonal index reflecting the lowest distance. The class distance between a given class and the reference class, from small to large, is indicated in ascending weight in each row. Moreover, we generate 36 LCA distance matrices from pretrained models on ImageNet. The results depicted in Figure 10 and Table 6 show an intermediate correlation between the in-domain LCA of the source model and the generalization of the linear probe model. They also indicate that a model's generalization could be modified by enforcing different inter-class distances, with limited changes to in-domain accuracy. Our future work will continue to explore the relationship between inter-class distance in pretrained models and their generalization.

| | ImageNet | | ImageNetv2 | | ImageNet-S | | ImageNet-R | | ImageNet-A | | ObjectNet | |
| --- | --- | --- | --- | --- | --- | --- | --- | --- | --- | --- | --- | --- |
| | PEA | SPE | PEA | SPE | PEA | SPE | PEA | SPE | PEA | SPE | PEA | SPE |
| LCA ->Hierarchy Linear Prob | 0.672 | 0.462 | 0.712 | 0.466 | 0.719 | 0.625 | 0.799 | 0.733 | 0.640 | 0.526 | 0.622 | 0.424 |

Table 6: **Correlation measurement between LCA matrix and In-domain LCA on ResNet18.** Following the algorithm of K-Means Clustering, we construct 36 LCA distance matrices (class hierarchies) from different pretrained models on ImageNet. We then use the LCA distance matrices as soft labels to guide linear probing on ResNet18 features. The table indicates the relationship between the In-domain LCA of the pretrained model and the out-of-distribution (OOD) accuracy on the linear probe model using the corresponding LCA distance matrix. The result is calculated from the average of three random seeds. Visualization is shown in Figure 10.

### F.5 Hyperparameters and Computational Resources

In the linear probing experiment, we chose hyperparameters based on the task at hand. The learning rate was set to 0.001, batch size=1024. We used the AdamW optimizer with a weight decay and a

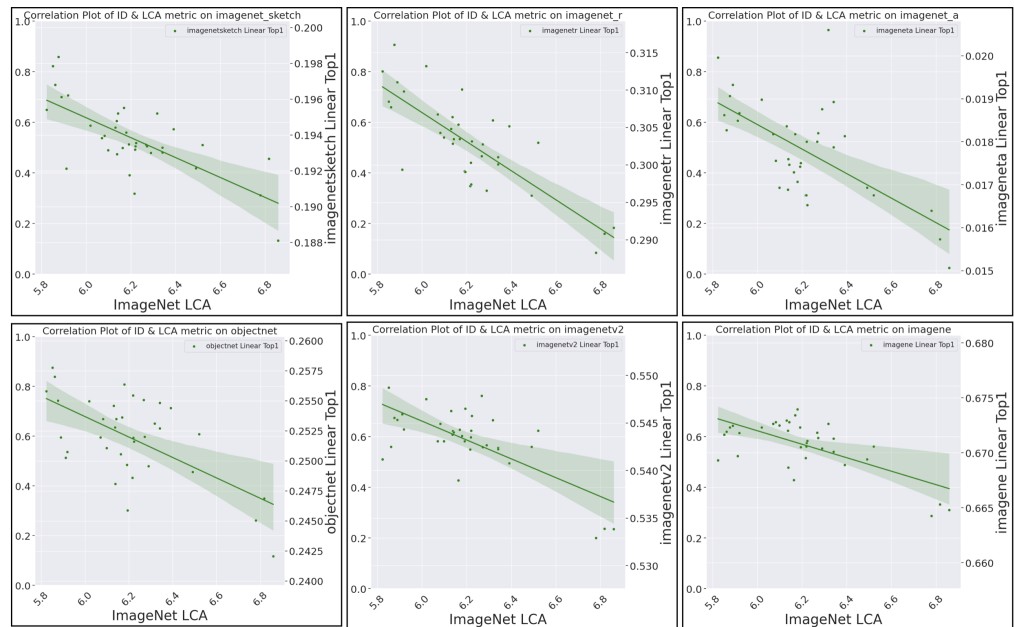

Figure 10: **Coorelation measurement between LCA matrix and In domain LCA on ResNet18.** Visualization on result in Tab 6. Plot shows an intermediate correlation between the two variable. If necessary, please find png of this image in supplementary for better legibility.

cosine learning rate scheduler with a warm-up iteration. The warm-up type was set to 'linear' with a warm-up learning rate of 1e-5. The experiment was run for 50 epochs.

For our computational resources, we utilized a single NVIDIA GeForce GTX 1080 Ti GPU.

# G  SUPPLEMENTARY RESULT

## G.1  IMPROVING GENERALIZATION BY CLASS TAXONOMY ALIGNMENT WITH PROMPT ENGINEERING

In this section, we present result of improving model Generalization by Taxonomy Integration in Vision-Language Models.

For vision-language models, we can easily incorporate taxonomy-specific knowledge by providing in-context information during zero-shot evaluation. Naturally, the WordNet (Miller, 1995) hierarchy implies the inter-class distance in data generation. For instance, adjacency in the hierarchy suggests that 'dalmatian' and 'husky' are semantically very close since both classes are derived from the same parent node 'dog'.

We present the results with CLIP-vit32 (Radford et al., 2021) in Tab 7. In an experiment to validate our proposal, we explicitly incorporated hierarchical taxonomy relationships into the prompt for zero-shot VLM prediction. We designed the prompt as **'A, which is a type of B, which is a type of C'** to inform the model to make predictions that align with the correct taxonomy. In addition, we included two ablation comparisons to show cases when 1) the correct taxonomy path is given, but the model is not informed of relationship between class names **(Stack Parent)**; and 2) the model is explicitly informed that a hierarchical 'is-a' relationship exists between class name, but the incorrect taxonomy relationship randomly sample from tree **(Shuffle Parent)** is provided. Our results demonstrate that only informing the model of the correct taxonomy and their hierarchical relationships can improve generalization. This is evidenced by improvements in Top-1 accuracy, ELCAD, and test-time cross-entropy across all datasets for all tested models.

| Model | ImageNet Top1 | ImageNet Test CE | ImageNet ELCA | ImageNetv2 Top1 | ImageNetv2 Test CE | ImageNetv2 ELCA | ImageNet-S Top1 | ImageNet-S Test CE | ImageNet-S ELCA | ImageNet-R Top1 | ImageNet-R Test CE | ImageNet-R ELCA | ImageNet-A Top1 | ImageNet-A Test CE | ImageNet-A ELCA | ObjectNet Top1 | ObjectNet Test CE | ObjectNet ELCA |
|---|---|---|---|---|---|---|---|---|---|---|---|---|---|---|---|---|---|---|
| Baseline | 0.589 | 1.635 | 9.322 | 0.517 | 2.014 | 9.384 | 0.379 | 2.817 | 9.378 | 0.667 | 1.348 | 8.790 | 0.294 | 3.098 | 9.358 | 0.394 | 2.631 | 8.576 |
| Stack Parent | 0.381 | 3.730 | 9.389 | 0.347 | 3.948 | 9.395 | 0.219 | 5.540 | 9.561 | 0.438 | 3.287 | 9.258 | 0.223 | 4.469 | 9.364 | 0.148 | 5.127 | 9.076 |
| Shuffle Parent | 0.483 | 2.236 | 9.679 | 0.432 | 2.586 | 9.696 | 0.329 | 3.251 | 9.718 | 0.557 | 1.919 | 9.281 | 0.236 | 3.532 | 9.586 | 0.329 | 3.067 | 8.785 |
| Taxonomy Parent | **0.626** | **1.457** | **9.102** | **0.553** | **1.824** | **9.165** | **0.419** | **2.544** | **9.319** | **0.685** | **1.279** | **8.658** | **0.319** | **2.839** | **9.171** | **0.431** | **2.433** | **8.515** |

Table 7: **Accuracy on OOD dataset by enforcing class taxonomy: Baseline**: *<dalmatian>*; **Stack Parent**: *<dalmatian, dog, animal>*; **Taxonomy Parent**:*<dalmatian, which is type of a dog, which is type of a animal >*; **Shuffle Parent**: *<dalmatian, which is type of a organism, which is type of a seabird>*; We have shown that only integrating the correct structure (inform the hierarchical 'is-a' relationship between class name) as well as correct value(valid taxonomy relationship) on WordNet could boost model performance and generalization.

## G.2 DOES IMAGENET LCA (TAXONOMY DISTANCE) REFLECT IMAGENET TOP 1 ACCURACY?

Here we present numeric result for discussion in the main paper. We challenge the common belief that LCA and Top-1 accuracy follow the same trend within the same dataset (Deng et al., 2009a; Bertinetto et al., 2020). As shown in 11 8, when including both VM and VLM zero-shot models, ImageNet and ImageNet-v2 show a weak correlation between LCA and Top-1 accuracy, while other semantically distinct OOD datasets exhibit a stronger relationship.

We hypothesize that it's due to overfitting of in domain feature. In our LCA-on-the-Line framework, we define model generalization(often noted as Top1 accuracy) related to the degree of alignment between model's prediction and the latent data generation process. In general case, LCA should be an unbiased measurement of such alignment. However, when we evaluate on In domain dataset(like ImageNet), and dataset that are visually similar to In domain dataset (like ImageNetv2), Top 1 accuracy fail to accurate reflect model's performance on general dataset(like naturally shrift semantic dataset) as it's 'inflated' from overfitting to specific training paradigms for in-distribution data. Thus model from different family (specifically VM and VLM) will overfit to their specific training mode as shown in two linear trend in plot of ImageNet/v2 in Fig 11, which weaken the correlation between LCA and Top1 accuracy.

For example, vision-only models often use cross-entropy to optimize class discrimination, which only trying to separate each class embedding cluster and fails to distribute embeddings in a semantically meaningful way. In contrast, vision-language models employ contrastive learning to align visual space with a well-regularized language embedding space, leading to semantically related classes being grouped closer together. This discrepancy in training paradigms means that Top 1 accuracy cannot accurately reflect the encoded decision process of class relationships for in domain dataset.

| Model | Group | ImageNet | | ImageNetv2 | | ImageNet-S | | ImageNet-R | | ImageNet-A | | ObjectNet | |
|---|---|---|---|---|---|---|---|---|---|---|---|---|---|
| | | *K^2* | *PEA* | *K^2* | *PEA* | *K^2* | *PEA* | *K^2* | *PEA* | *K^2* | *PEA* | *K^2* | *PEA* |
| | ALL | 0.237 | 0.488 | 0.259 | 0.509 | **0.838** | **0.915** | **0.749** | **0.865** | **0.869** | **0.932** | **0.672** | **0.820** |
| | | *KEN* | *SPE* | *KEN* | *SPE* | *KEN* | *SPE* | *KEN* | *SPE* | *KEN* | *SPE* | *KEN* | *SPE* |
| | | 0.293 | 0.302 | 0.298 | 0.380 | **0.828** | **0.937** | **0.600** | **0.795** | **0.813** | **0.948** | **0.727** | **0.901** |
| Top1->LCA | VLM | K^2 | PEA | K^2 | PEA | K^2 | PEA | K^2 | PEA | K^2 | PEA | K^2 | PEA |
| | | **0.934** | **0.966** | **0.886** | **0.941** | **0.922** | **0.960** | **0.889** | **0.943** | **0.792** | **0.890** | 0.570 | 0.755 |
| | | KEN | SPE | KEN | SPE | KEN | SPE | KEN | SPE | KEN | SPE | KEN | SPE |
| | | **0.848** | **0.955** | **0.684** | **0.853** | **0.867** | **0.959** | **0.686** | **0.861** | **0.689** | **0.879** | 0.494 | **0.704** |
| | VM | K^2 | PEA | K^2 | PEA | K^2 | PEA | K^2 | PEA | K^2 | PEA | K^2 | PEA |
| | | **0.976** | **0.987** | **0.893** | **0.945** | **0.895** | **0.945** | 0.095 | 0.310 | **0.833** | **0.913** | **0.913** | **0.956** |
| | | KEN | SPE | KEN | SPE | KEN | SPE | KEN | SPE | KEN | SPE | KEN | SPE |
| | | **0.911** | **0.982** | **0.821** | **0.942** | **0.825** | **0.949** | 0.149 | 0.222 | **0.782** | **0.917** | **0.838** | **0.957** |

Table 8: **Correlation measurement between Top 1 and LCA** on 77 models across modality (37 VM and 40 VLM) on 6 datasets; For instance, Corr(ImageNet Top1 Acc, ImageNet LCA) or Corr(ImageNet-A Top1 Acc, ImageNet-A LCA); Follow Fig 11. We highlight strong correlation indications. We take the absolute value of all correlations for simplicity.

## G.3 RANKING MEASUREMENT OF LCA-ON-THE-LINE

Here we present the numeric result for ranking measures of *KEN (Kendall rank correlation coefficient)* and *SPE (Spearman rank-order correlation coefficient)* in comparision to common use Top1 In domain

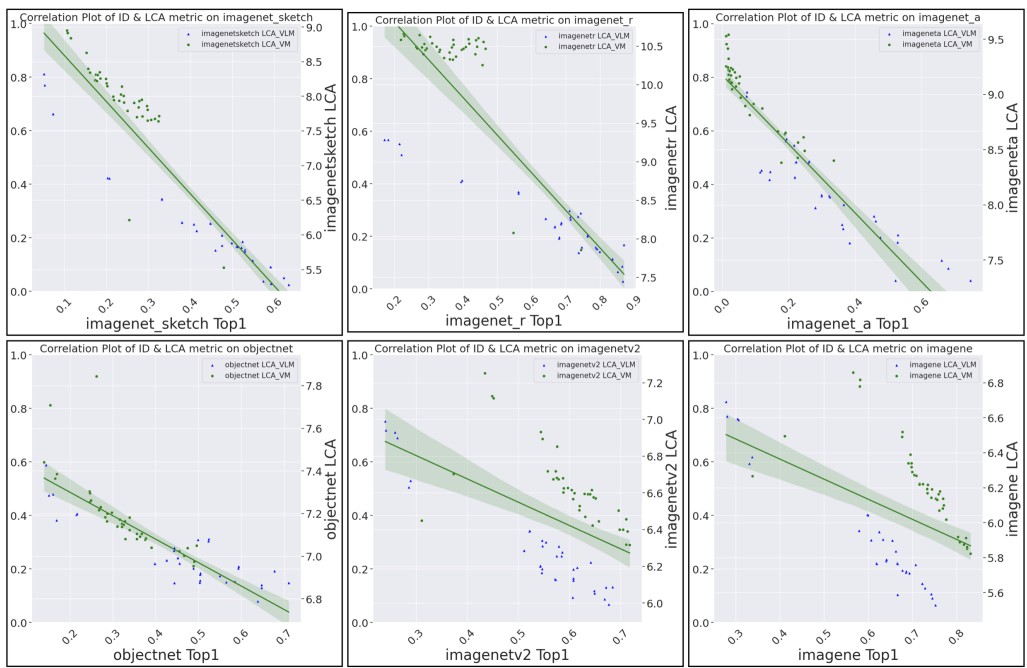

Figure 11: **Predicting LCA (VM+VLM, 75 models) on 6 ImageNet-variant Datasets** Following Tab 8. For each plot, the x-axis indicates dataset Top-1 accuracy, while the y-axis indicates LCA distance. From the plots, it is clear that ImageNet and ImageNet-v2 do not show a strong correlation between LCA and Top-1 accuracy, while other semantically distinct OOD datasets exhibit a stronger relationship. Additionally, this challenges the common belief that in-domain Top-1 accuracy and LCA distance follow the same order (Deng et al., 2009b; Bertinetto et al., 2020). Please refer to the discussion for further details. If necessary, please find png of this image in supplementary for better legibility.

accuracy in 9. Equalevently, in domain LCA measure present strong result in both preserving linearity and ranking.

| | Element | | ImageNetv2 | | ImageNet-S | | ImageNet-R | | ImageNet-A | | ObjectNet | |
|---|---|---|---|---|---|---|---|---|---|---|---|---|
| | ID | OOD | KEN | SPE | KEN | SPE | KEN | SPE | KEN | SPE | KEN | SPE |
| ALL | Top1 | Top1 | **0.840** | **0.947** | 0.170 | 0.092 | 0.146 | 0.042 | 0.068 | 0.037 | 0.317 | 0.339 |
| | LCA | Top1 | 0.421 | 0.517 | **0.828** | **0.937** | **0.761** | **0.911** | **0.813** | **0.948** | **0.867** | **0.967** |
| | Top1 | Top5 | **0.672** | **0.818** | 0.151 | 0.059 | 0.134 | 0.004 | 0.108 | 0.021 | 0.279 | 0.297 |
| | LCA | Top5 | 0.571 | 0.729 | **0.843** | **0.948** | **0.752** | **0.897** | **0.817** | **0.947** | **0.861** | **0.966** |
| VLM | Top1 | Top1 | **0.971** | **0.997** | 0.840 | 0.936 | **0.864** | **0.943** | 0.753 | 0.915 | **0.905** | **0.982** |
| | LCA | Top1 | 0.882 | 0.972 | **0.867** | **0.959** | 0.762 | 0.886 | **0.800** | **0.942** | 0.870 | 0.972 |
| | Top1 | Top5 | 0.908 | 0.980 | 0.848 | **0.951** | **0.882** | **0.959** | 0.753 | 0.910 | **0.842** | **0.964** |
| | LCA | Top5 | 0.900 | **0.981** | **0.856** | 0.950 | 0.775 | 0.907 | **0.794** | **0.943** | 0.829 | 0.955 |
| VM | Top1 | Top1 | **0.948** | **0.993** | 0.771 | 0.901 | **0.743** | **0.887** | 0.735 | 0.877 | 0.822 | 0.927 |
| | LCA | Top1 | 0.910 | 0.981 | **0.825** | **0.949** | 0.705 | 0.862 | **0.782** | **0.920** | **0.838** | **0.957** |
| | Top1 | Top5 | **0.939** | **0.992** | 0.752 | 0.894 | **0.758** | **0.901** | 0.818 | 0.941 | 0.815 | 0.920 |
| | LCA | Top5 | 0.894 | 0.977 | **0.832** | **0.951** | 0.707 | 0.871 | **0.824** | **0.939** | 0.846 | 0.958 |

Table 9: **Ranking measurement of ID LCA/Top1 with OOD Top1/Top5** on 75 models across modality(36 VM and 39 VLM); As shown in the 'ALL grouping', LCA shows a much better result in preserve in model relative ranking to model OOD performance on all OOD datasets (with the exception of ImageNet-v2), which indicate the superiority for model selection.

