# OpenReview forum: "LCA-on-the-Line: Benchmarking Out-of-Distribution Generalization with Class Taxonomies"
_ICLR.cc/2024/Conference — Submitted to ICLR 2024_

### Official Review · Reviewer_CuPJ · 2023-10-17

**Soundness:** 2 fair
**Presentation:** 1 poor
**Contribution:** 2 fair
**Rating:** 3
**Confidence:** 4

**Summary:**

This paper focuses on addressing the challenge of assessing model generalization under out-of-distribution conditions. They reintroduce the Least Common Ancestor (LCA) distance. Particularly, they utilize the LCA to measure the taxonomic distance between labels and predictions, presenting it as a benchmark for model generalization. In the experiments, the proposed method is evaluated on multiple datasets.

**Strengths:**

It is interesting to address the challenge of assessing model generalization under out-of-distribution conditions.

**Weaknesses:**

1. The Introduction Section is not clear. The authors indicate that most methods involve modeling correlations with in-domain accuracy or agreement. And many studies evaluate generalization on OOD datasets that feature limited visual shifts. These interpretations are very unclear. I am not clear the concrete meaning. I recommend the authors modify their paper carefully.

2. The authors indicate that to address the analyzed issues, they introduce a method to benchmark model generalization, i.e., using the taxonomy loss. Firstly, the authors do not interpret whether the research is meaningful clearly. Secondly, the authors do not sufficiently introduce the advantages of the taxonomy loss. I recommend the authors draw a figure to clearly describe the motivation.

3. In Table 1, the evaluated methods are somewhat old. The authors should verify the effectiveness of the proposed method on more state-of-the-art methods, e.g., the works from CVPR 2023, ICLR 2023. Meanwhile, the experiments are somewhat unclear. The authors only evaluate the classification performance. I recommend the authors evaluate the proposed method on other tasks, e.g., object detection and semantic segmentation. Finally, for Fig. 1, the authors should give more interpretations.

**Questions:**

See Weakness.

---

> ### Author Response · Authors · 2023-11-13
>
> We thanks reviewer for appreciate the interestingness of our paper and raising concerns about the clarity of the presentation, motivation and relevancy toward other field.
>
> **Q: The Introduction Section is not clear, especially about the description of ‘most methods involve modeling correlations with in-domain accuracy or agreement. And many studies evaluate generalization on OOD datasets that feature limited visual shifts.’ These interpretations are very unclear. I am not clear the concrete meaning.**
>
> - Please let us know if there’s any other place remain unclear and we are happy to take the advice to rephrase those sentences in paper at the end of the discussion!
>
> - We want to echo that our main motivation is to establish ‘a unified benchmarking metric robustly applicable across both VM and VLM, to assess model generalization (use in domain measurement to predict model’s OOD performance).
>
> - **most methods involve modeling correlations with in-domain accuracy or agreement.** ->
> Prior work, notable, ‘accuracy-on-the-line’[1] and ‘agreement-on-the-line’[2], are two highly influenced papers in the field. The former is the first work to establish a strong baseline that the models with better ID accuracy are likely to have better OOD accuracy, while the latter adopted model ensemble and argued that OOD agreement between the predictions of any two pairs of neural networks shows linear correlation with their ID agreement. Those two works setup important base tune for followup works. However, those two work did not explicitly evaluated models of different modality(VM & VLM) in comparison. Our results shows that those two well-acknowledged measurement are less robust when comparing models under different settings.
>
> - **many studies evaluate generalization on OOD datasets that feature limited visual shifts.** -> Many prior work( as referred in the introduction), where they found strong linear relationship between ID and OOD accuracy, are adopting OOD dataset that are visually similar to ID dataset, for instance, ImageNet-v2 or ImageNet-C. Our work argue that those dataset, did’t reflect more realistic distribution shrift, make then more like ‘another pool of ID dataset rather than realistic OOD dataset’(please also refer to section 4 for discussion.)  We instead focus on a more realistic distribution shift involving semantic concept shrift(like cartoon Dog and animal Dog) or orientation variance(like fallen chair in ObjectNet). We shows that LCA as a benchmark could predict model’s generalization on those realistic OOD dataset across both VM and VLMs..
>
> **Q: Firstly, the authors do not interpret whether the research is meaningful clearly. Secondly, the authors do not sufficiently introduce the advantages of the taxonomy loss.**
>
> - We thanks reviewer for raising concerns of our motivation. We would like to echo here to the main motivation of our work.
> - 1. Since the early year of ImageNet (2011) datasets,  LCA distance was proposed but largely dismissed as it was widely believed to follow the same ordering as Top 1 accuracy. This metric have been revisit many times in follow-up papers of hierarchical classification but the concept that ‘LCA don’t provide much information besides Top1 accuracy’ are still largely accepted. In our work, we want to challenge this prevailing notion and shows that LCA could provide strong signal to model’s feature learning in the era of Vision Language Model. This could provide some pointer of why VLM with lower ID accuracy are having a higher OOD compared to SOTA VM model.
> - 2. We then shows how LCA could solve the problem that haven’t been solved in prior work. In summary, prior work predicting model generalization are using 1.(VLM or VM) and 2.(visually similar dataset like ImageNet-v2 as OOD for ImageNet). This even include two highly influenced work [1][2]. We targeting on a more realistic setup of 1(VLM and VM) and 2. Various realistic OOD datasets like ImageNet-R/S/A/ObjectNet.
> - 3. We then provide some exploration on how‘LCA-on-the-line’ could be used to improve model’s generalization by applying taxonomy loss. This is introduced in 3.4 ENHANCING GENERALIZATION THROUGH CLASS TAXONOMY ALIGNMENT. We consider applying taxonomy loss similar to ‘alignment to human knowledge’ during model training.
> - For adding a figure indicating motivation as reviewer suggestion, Figure 2 is well motivated by our main motivation.

---

> ### Author Response · Authors · 2023-11-13
>
> **Q: In Table 1, the evaluated methods are somewhat old. The authors should verify the effectiveness of the proposed method on more state-of-the-art methods, e.g., the works from CVPR 2023, ICLR 2023.**
>
> - We want to point out that in our work, the main motivation is to identify a problem for which prior arts is suffering catastrophically( correlation of [1] drop from 0.9->0.01 in Table2) , and how LCA could still be robust under this settings. To the best of our knowledge, the only work that trying to address similar problem is one concurrent work, (Shi et al., 2023)[3](NeurIPS 2023), which argued to use multiple OOD test set as model generalization indicator. We have discussed this work in the introduction for sharing similar motivation.
>
> **Q: The authors only evaluate the classification performance. I recommend the authors evaluate the proposed method on other tasks, e.g., object detection and semantic segmentation.**
>
> - We thanks reviewer for suggesting promising investigation towards other field. We want to point that object detection and semantic segmentation, specifically, involved both localization and classification uncertainty. We believe the discussion of localization uncertainty are out of scope of this paper and could become a potential impactful future work that require full investigation. LCA mainly deal with semantic understanding of feature learning thus only related to classification uncertainty. We thanks reviewers again for the suggestion and will include this in the future work section.
>
> **Q: For Fig. 1, the authors should give more interpretations.**
> - We thanks reviewer for identifying clarity of Fig. 1. We will make sure to rephrase it to make it easier to understand. In a high level, Fig 1 shows that prior art fail to maintain a straight correlation line when involved both VM and VLM, while LCA as measurement achieved a straight line on ImageNet S/R/A/O. This is aligning with numeric value in table 2. Please feel free to provide more suggestion where you feel appropriate!
>
> [1] John P Miller, Rohan Taori, Aditi Raghunathan, Shiori Sagawa, Pang Wei Koh, Vaishaal Shankar, Percy Liang, Yair Carmon, and Ludwig Schmidt. Accuracy on the line: on the strong correlation between out-of-distribution and in-distribution generalization. In International Conference on Machine Learning, pp. 7721–7735. PMLR, 2021.
>
> [2] Christina Baek, Yiding Jiang, Aditi Raghunathan, and J Zico Kolter. Agreement-on-the-line: Predicting the performance of neural networks under distribution shift. Advances in Neural Information Processing Systems, 35:19274–19289, 2022.
>
> [3] Zhouxing Shi, Nicholas Carlini, Ananth Balashankar, Ludwig Schmidt, Cho-Jui Hsieh, Alex Beutel, and Yao Qin. Effective robustness against natural distribution shifts for models with different training data. arXiv preprint arXiv:2302.01381, 2023.

---

> ### Author Response · Authors · 2023-11-23
>
> Dear Reviewer
> We had modified our paper in detail, especially on introduction based on your suggestions. We have also include 7 more plots to improve readability. Please refer to our updated draft for detail. Please also refer to general comment we have for all reviewers, which include summary of our change and detail summary of our work. If you think the presentation have been improved and appreciate our change in the updated submission, please raise rating score. We understand it's approaching to the end of reviewer-author discussion, please review our updates for your decision in the AC-reviewer discussion stage. Thank you!

---

### Official Review · Reviewer_r8af · 2023-10-31

**Soundness:** 3 good
**Presentation:** 3 good
**Contribution:** 2 fair
**Rating:** 5
**Confidence:** 3

**Summary:**

In this paper, the Least Common Ancestor (LCA) distance via the WordNet hierarchy is employed to measure the taxonomic distance between labels and predictions, utilizing it as a benchmark for model generalization. Extensive experiments are performed on model evaluation, including vision-only and vision-language models on natural distribution shift datasets. A strong linear correlation is observed between in-domain ImageNet LCA scores and out-of-domain (OOD) Top1 performance across many variants of ImageNet.

**Strengths:**

A thorough experimental analysis of the relationship between Out-of-Distribution (OOD) detection performance and Least Common Ancestor (LCA) distance for image classifiers is provided in this paper. The following are key strengths of the work:

(1) Experiments are conducted on a diverse set of neural network architectures, including ResNet, VGG, EfficientNet, Vision Transformer (ViT), and Vision-Language Models (VLMs), enabling conclusions that potentially generalize across problem domains. The rigor and comprehensiveness of the methodology are underscored by the scale of experiments, involving up to 75 network variants.

(2) A major point about this submission is the analysis that demonstrates a correlation between OOD detection performance and LCA distances in the classifier. For instance, it is quantitatively shown how images from OOD datasets with more distant LCA relationships to the training data tend to be easier to detect as anomalies. The intuitive justification provided is that greater separation between the semantics of the origin dataset and OOD dataset in the hierarchical LCA structure leads to more separable distributions.

**Weaknesses:**

Major:

(1) The paper does not evaluate multiple OOD scoring methods like energy scores[1], ODIN[2], Mahalanobis distance[3], and ReAct[4], which would have provided insights into the validity of the key conclusions across different anomaly scoring approaches. The interaction between the choice of scoring method and LCA distance remains unclear. Understanding how these scoring methods affect OOD performance from the perspective of the LCA distance is important. It should be noted that the calculation of this LCA distance is limited to variants of ImageNet considering the WordNet hierarchy used.

(2) The claim is made that the findings offer "invaluable insights and actionable techniques" to enhance robustness and generalization. However, no concrete solutions leveraging the LCA distance are proposed or analyzed. Further theoretical or empirical analysis that establishes a connection between these insights and improved generalization would be beneficial.


Minor:

“Given two classes, y (the ground truth class) and y′, we define the LCA distance according to (Bertinetto et al., 2020) as lcad(y′, y) := f(y) − f(lca(y, y′), where f(y) ≥ f(lca(y, y′) and
lca((y′, y) denotes…” All formulations lose the right brackets.

“As highlighted in Fig 1 (indicated in red), when adhering to ’accuracy on the line’,” Wrong quotes.

In summary, the rigorous experiments and novel analysis of OOD detection vs. LCA distance are valuable contributions, but additional evaluation of alternative scoring methods and practical applications of the findings could further strengthen the work.


[1] Liu, Weitang, et al. "Energy-based out-of-distribution detection." Advances in neural information processing systems 33 (2020): 21464-21475.

[2] Liang, Shiyu, Yixuan Li, and Rayadurgam Srikant. "Enhancing the reliability of out-of-distribution image detection in neural networks." ICLR 2018.

[3] Ren, Jie, et al. "A simple fix to mahalanobis distance for improving near-ood detection." arXiv preprint arXiv:2106.09022 (2021).

[4] Sun, Yiyou, Chuan Guo, and Yixuan Li. "React: Out-of-distribution detection with rectified activations." Advances in Neural Information Processing Systems 34 (2021): 144-157.

**Questions:**

Please see the weaknesses above.

---

> ### Author Response · Authors · 2023-11-13
>
> We thanks reviewer for appreciating our rigor and comprehensive experiment.
>
> For the (2) in strengths, we believe there’s some misunderstandings about our experiment setup. **Our paper didn’t try to target OOD detection or anomalies detection(separate ID data from and OOD data, binary classification), but to predict whether a model could perform better on OOD dataset by only looking at its in-domain measurement**(correlation of model top1 accuracy between ID and OOD). There’s a difference in the problem setup. We could like to point reviewer to Table 2 for reference.
> - **it is quantitatively shown how images from OOD datasets with more distant LCA relationships to the training data tend to be easier to detect as anomalies.** -> We actually trying to show model that have a lower LCA distance on ID will have a higher top 1 accuracy on OOD.
>
> - **The intuitive justification provided is that greater separation between the semantics of the origin dataset and OOD dataset in the hierarchical LCA structure leads to more separable distributions.** -> Please refer to our answer below about assumption.
>
> **Q: Understanding how these scoring methods affect OOD performance from the perspective of the LCA distance is important. It should be noted that the calculation of this LCA distance is limited to variants of ImageNet considering the WordNet hierarchy used.**:
>
> - We thanks reviewer for point to scoring function. We will make sure to cite those work in our revision. Here’s our interpretation of suggested work and please feel free to suggest more!
>
> - The suggested works are targeting out-of-distribution detection, which trying to tell which data are OOD and which data have been seemed during training. This is a different task from our paper. Pleaser refer to answer above.
> - Different from our focus, the main targets of those methods is to **separate feature between ID and OOD data**, which our focus is more detail, to identify within ID feature, how does model learned to separate class semantically? Under the assumption that **ID and OOD data should lands on a share feature distribution, rather than two separate ones**. Assuming two separate distribution is what being assumed in OOD detection. However, we still believe this is an interesting difference worth to mention in the discussion as related work and will make sure to include citation of those work.
>
> **Q: It should be noted that the calculation of this LCA distance is limited to variants of ImageNet considering the WordNet hierarchy used.**
> - We want to point reviewer to section 3.4. In order to extends our metric beyond ImageNet and WordNet, we proposed to infer pseudo latent class taxonomy using K-Means Clustering as a replacement to replying on WordNet hierarchy. Our result in table 4 shows that our latent taxonomy are equally robust to the experiment in Table 2 among 75 different latent taxonomy.
>
> **Q: However, no concrete solutions leveraging the LCA distance are proposed or analyzed.**
> - We want to point reviewer to section 3.4 and F.1 in appendix. In section 4 we have adopt our idea LCA-on-the-line as a taxonomy loss,  and explore how it could used to improve model’s generalization. Similarly in F.1, we shows that how this idea could be used on prompting. We have also analyzed the underlying connections between why LCA are suitable generalization metric in 3.3. Please feel free to ask more question where you consider less clear.
>
>
> **Q: Minor error:**
> - Thanks for identifying typo in brackets, we will update accordingly.
> - For “As highlighted in Fig 1 (indicated in red), when adhering to ’accuracy on the line’,” Wrong quotes.”, sorry we don’t understand. Could you provide more detail on why this is wrong quotes? Thanks!

---

> ### Author Response · Authors · 2023-11-21
> **Citation updated in introduction.**
>
> Dear reviewer
> We have updated our submission by including reference of works in OOD detection and proper citation in the introduction. Please let us know with other questions！

---

### Official Review · Reviewer_5fCB · 2023-11-03

**Soundness:** 4 excellent
**Presentation:** 2 fair
**Contribution:** 4 excellent
**Rating:** 8
**Confidence:** 3

**Summary:**

This paper focuses on evaluating the Out-of-Distribution (OOD) generalization by using Least Common Ancestor (LCA) distance based on the WordNet hierarchy. LCA measures the taxonomic distance between labels and predictions, and the paper shows that LCA is a better measure of OOD generalization than top-1 accuracy (when both LCA and top-1 accuracy are computed on in-domain data). Intuitively, LCA is able to better evaluate how well a model has learned semantic knowledge of the classes (since lower LCA indicates that the model’s wrong predictions are semantically closer to the true class). This enables LCA to be a better measure of OOD performance compared to top-1 accuracy which only considers whether the prediction is correct or not. Specifically, they test 75 different models (including both vision models and vision-language models) and find a linear correlation between ImageNet LCA and OOD accuracy across 4 standard ImageNet-OOD datasets. They also use the pairwise LCA between classes as soft labels for linear probing over pretrained models, and find that OOD performance can be improved at the cost of in-domain performance.

**Strengths:**

* The idea of using LCA for evaluating OOD generalization is simple, intuitive, well-motivated, and effective.

* The paper is fairly well-written (except some typos which can be fixed).

* The experimental analyses are quite extensive. The contributions of this work will likely be quite significant for industry applications where predicting OOD performance ahead of deployment tends to be important.

**Weaknesses:**

* Explanation of LCA is complicated
    * A visual illustration of LCA computation with a small part of the WordNet hierarchy and 2-3 example pairs of classes would help readers to quickly and better understand the LCA distance measure.
    * Maybe a figure in the main paper like Fig. 3 (in Suppl.) but with some actual classes and example LCA values for a few pairs of classes.

* Fig. 1 is very difficult to read and understand
    * The legend is too small. It would be better to show only top-1 here (first row) with larger font sizes (match caption size, roughly) and show the full figure in supplementary.

* Implementation of inferred class taxonomy is difficult to understand
    * It is unclear what "establishing the cluster level where both classes share the same cluster as the height of LCA" means. Please clarify and try to simplify it.
    * A figure illustrating the method would be ideal to help readers understand it better.

**Questions:**

* Please see the weaknesses section.

* Minor comments
    * Abstract (second paragraph) has a typo: “Beside” → “Besides”.
    * Introduction (second paragraph): “effective robustness(Taori et al., 2020)” space needed between robustness and the citation.
    * Above the contributions list, typo: “in measure model’s semantic awareness” → “in measuring a model’s semantic awareness”.
    * Sec. 2 (second paragraph): extra or less brackets in three equations in this paragraph.
    * In many places in the paper, quotes are used incorrectly in LaTeX. Please use ` ' in LaTeX (i.e. backtick and quote instead of both quotes).
    * Paragraph below Table 3 has a typo: “As illustrated in Table3” → “As illustrated in Table 3”, i.e. add space.
    * Sec. 3.3 (last paragraph) has a typo: “natural image(ImageNet)” → “natural image (ImageNet)”, i.e. add space.

---

> ### Author Response · Authors · 2023-11-13
>
> We highly appreciate reviewer for identifying our work as ‘simple, intuitive, well-motivated, and effective’, well-written, extensiveness of our work and potential significant impact. We also thanks reviewer for proving detail for improvement.
>
> **Q: Explanation of LCA is complicated**
> - Thanks for the suggestion! We will point reader more directly to plot in appendix, and provide an actual class example in appendix plot.
> A simple intuition would be,
> animal have children of cat and dog;
> Dog have children of chihuahua and Husky;
> Etc.
>
> **Q: Fig. 1 is very difficult to read and understand**
> - Thanks for identify the lack of legibility of Fig1. We have included original png of Figure 1 in the supplementary zip folder for reference. We will also make sure to enlarge the text in our revision.
>
> **Q: Implementation of inferred class taxonomy is difficult to understand**
> - Thanks for the suggestion, we will make sure to include a algorithm pseudocode in our revision.
> - **establishing the cluster level where both classes share the same cluster as the height of LCA** -> Similar to idea of LCA in FIgure3 in appendix, we form a hierarchical tree by using different number of clustering in each layers. For instance, in level 1 we have all the class, in level 2 we have two tree node(result from K-mean of 2 cluster), in level 3 we have 4 tree node(result from K-mean of 4 cluster), etc.
>
> **Q: Minor typo/grammar issue**
> - We appreciate for noticing those minor issue and we have corrected them all in the revision.

---

> ### Comment · Reviewer_5fCB · 2023-11-23
> **Response to Authors**
>
> I thank the authors for their efforts during the discussion period. The response has mostly addressed my concerns and I keep my already positive score of 8: accept, good paper.

---

> ### Author Response · Authors · 2023-11-23
>
> We are glad that our reply had addressed reviewer’s concern! We hope reviewer also consider raising score on presentation if you believe our revision have improved the clarity of paper presentation.
> Thanks!

---

### Author Response · Authors · 2023-11-20
**Looking forward to hear from reviewer!**

We highly appreciate the feedback from reviewer on our work, and we have go over each point carefully to provide a reply. Please let us know if the reply address the concern. We would love to incorporate your suggestions into our refined version of the work!

**Below please find attach a comprehensive summary of our work. Hope this is useful for AC and reviewer!**

In this study, we focus on evaluating how well models generalize to unseen, out-of-distribution (OOD) datasets. Specifically, we aim to predict a model's OOD performance, based on its performance in a familiar, in-domain setting.

Previous research in this area has primarily focused on measuring model performance on OOD datasets with limited visual shifts.

Or have not adequately compared vision models trained on datasets like ImageNet, with vision-language models trained on datasets like LAION.

To address these gaps, we introduce 'LCA-on-the-line.' This is the first approach to uniformly measure model robustness across different model modalities and OOD datasets with significant distribution shifts.

Our approach is grounded in the understanding of how models learn. During training, models learn to classify by creating decision boundaries and extracting distinguishing features between classes. A model's ability to generalize to OOD datasets depends on the transferability of these learned features during training. This brings us to the concept of spurious correlations. For example, a model might mistakenly associate grass with ostriches if they frequently appear together in training, and expecting the same association during testing.

We hypothesize that only semantic features that align with human understanding of object definitions are universally transferable to any OOD dataset. For example, a model that learns to associate ostriches with features like 'long legs' and 'long neck', which are more transferable to OOD datasets, will likely predict classes semantically closer to the ground truth, such as flamingos. In contrast, a model influenced by spurious correlations by falsely associate ostrich with grass, might predict a semantically distant class, like an Jaguars, which are also appear often on grass.

Our method involves measuring a model's generalization based on its in-domain semantic severity of mistake. We use the 'Least Common Ancestor' (LCA) distance, which is the distance between the model's prediction and the ground truth class in a predefined taxonomy hierarchy, like WordNet.

We tested our hypothesis using 75 models with varying settings, including vision models trained on ImageNet and vision-language models trained on LAION. We used ImageNet as the in-domain dataset and four OOD datasets with significant distribution shifts.

Our results show that the in-domain LCA distance on the ImageNet test set correlates robustly and linearly with accuracy across all OOD test sets. This finding challenges previous studies using in-domain accuracy that failed to establish a unified measurement for models trained on different datasets.

For dataset without a predefined hierarchy, we construct a latent hierarchy by apply K-mean to group classes into different node at each level of the tree, achieved robust performance similar to WordNet hierarchy.

Additionally, our work explores how aligning model predictions with class hierarchies during training can improve generalization.

Lastly, our research challenges the long-held belief that LCA distance offers no additional insights beyond top-1 accuracy since ImageNet dataset. We provide a compelling alternative to understand why vision-language models with lower in-domain accuracy might generalize better to OOD datasets than vision models.

---

### Author Response · Authors · 2023-11-21
**Significant update to submission by updating 7 plots.**

Dear reviewer, please refer to our updated submission for reference. We have updated 7 plots as suggested in your valuable review to improve readability. We have also address other minor issue in our update. Please take a look and see if the updated plot address your concern. We are looking forward to hear from you!

Reviewer 5fCB:
- (Figure 2) We have include a plot visualization LCA score with actual class in our main paper.
- (Figure 7) We have also include a plot illustrating K-mean clustering.

Reviewer r8af:
- (Figure 1) We have updated an teaser figure in introduction to address the problem we are targeting.

Reviewer CuPJ:
- (Figure 8) We have updated a plot in appendix illustrate our setting compare to prior work.
- (Figure 5,6)  We have also include 2 plots in section 3.3 to explain our motivation and significant of the work.
- (Figure 4) We have also include one plot beside Fig. 1 to give more interpretation.

---

### Author Response · Authors · 2023-11-22

Dear reviewers

It's the last day in the rebuttal discussion period. Please take a look of our reply to see if it adequately address your concerns. Please also provide any last minute concern and we will try to reply within the last day / or in future revision if no enough time. We have updated 7 plots to address concerns from reviewers, and had also modified texts from abstract, introduction, explanation in experiment, and major text in motivation of adopting LCA in 3.3 THE SUITABILITY OF LCA AS A BENCHMARK FOR MODEL GENERALIZATION. If you feel our answer resolved your concern, please consider raising score, as well as score in Soundness/Presentation/Contribution. We are also more than happy to include any missing citations, or add additional discussion/explanation to our submission.

---

### Author Response · Authors · 2023-11-23
**Have updated another draft of our paper.**

Dear reviewer & AC
We have uploaded another version of our submission just now. We have further made many modification to texts and swap order of some paragraphs for readability. Please refer to this draft for your final decision. Thanks!

---

### Meta-Review · Area_Chair_bpcJ · 2023-12-14

**Metareview:**

This paper introduces the Least Common Ancestor (LCA) distance based on the WordNet hierarchy as a benchmark for evaluating model generalization under out-of-distribution conditions. LCA measures taxonomic distance, offering a superior assessment of out-of-distribution (OOD) generalization compared to top-1 accuracy. The study involves testing 75 models, including vision and vision-language models, revealing a linear correlation between ImageNet LCA and OOD accuracy across various datasets. Additionally, using pairwise LCA between classes as soft labels enhances OOD performance at the expense of in-domain accuracy. The proposed method addresses the challenge of assessing model generalization in diverse scenarios, providing insights into semantic knowledge learning and OOD performance improvement. Strengths include a well-motivated and effective concept, extensive experimental analyses with up to 75 network variants, and potential significant contributions for industry applications. However, weaknesses include a complicated explanation of LCA, unclear visual representations (e.g., Fig. 1), and a lack of clarity in the Introduction. The authors should consider simplifying explanations, providing visual aids, and clarifying their motivations. Additionally, the evaluated methods in Table 1 are somewhat outdated, and the paper should include more recent approaches. The proposed method's effectiveness should be verified against state-of-the-art models, and evaluations on other tasks like object detection and semantic segmentation are recommended. The paper also lacks comparisons with various OOD scoring methods, limiting insights into the LCA distance's validity across different anomaly scoring approaches. Lastly, while the paper claims to offer actionable techniques, it lacks concrete solutions or in-depth analyses leveraging the LCA distance for enhancing robustness and generalization. Therefore, I have determined that this paper does not meet the acceptance criteria and recommend rejecting it.

**Justification For Why Not Higher Score:**

This paper lacks innovation and rigor, with predominantly negative feedback from reviewers and significant disagreements. Therefore, it is not suitable for a recommendation for acceptance.

**Justification For Why Not Lower Score:**

N/A

---

### Decision · Program_Chairs · 2024-01-16

Reject